# LEAD: Large Foundation Model for EEG-Based Alzheimer's Disease Detection

## Abstract

Electroencephalography (EEG) provides a non-invasive, highly accessible, and cost-effective approach for detecting Alzheimer's disease (AD). However, existing methods, whether based on handcrafted feature engineering or standard deep learning, face two major challenges: 1) the lack of large-scale EEG-AD datasets for robust representation learning, and 2) the absence of a dedicated deep learning pipeline for subject-level detection, which is more clinically meaningful than the commonly used sample-level detection. To address these gaps, we have curated the world's largest EEG-AD corpus to date, comprising 2,255 subjects. Leveraging this unique data corpus, we propose LEAD, the first large-scale foundation model for EEG analysis in dementia. Our approach provides an innovative framework for subject-level AD detection, including: 1) a comprehensive preprocessing pipeline such as artifact removal, resampling, and filtering, and a newly proposed multi-scale segmentation strategy, 2) a subject-regularized spatio–temporal transformer trained with a novel subject-level cross-entropy loss and an indices group-shuffling algorithm, and 3) AD-guided contrastive pre-training. We pre-train on 12 datasets (3 AD-related and 9 non-AD) and fine-tune/test on 4 AD datasets. Compared with 10 baselines, LEAD consistently obtains superior subject-level detection performance under the challenging subject-independent cross-validation protocol. On the benchmark ADFTD dataset, our model achieves an impressive subject-level Sensitivity of 90.91% under the leave-one-subject-out (LOSO) setting. These results strongly validate the effectiveness of our method for real-world EEG-based AD detection. Source code: https://anonymous.4open.science/r/LEAD-3B51

## 1 Introduction

Alzheimer's Disease (AD) is the most common neurodegenerative disorder in the elderly, affecting 38 million individuals with $1.3 trillion annual financial cost Breijyeh & Karaman (2020); Masters et al. (2015); National Academies of Sciences, Engineering,

**Table 1: Comparison of different AD detection technologies.** CSF: Cerebrospinal Fluid.

| Technologies | Accessibility↑ | Safety↑ | Affordability↑ | Accuracy↑ |
|---|---|---|---|---|
| CSF Salvadó et al. (2024) | Low | Low | Low | High |
| Neuroimaging Ottoy et al. (2025) | High | High | Low | High |
| Blood Test Grande et al. (2025) | Low | Moderate | Moderate | Moderate |
| Neurological Exam Vöglein et al. (2023) | High | High | Moderate | Moderate |
| EEG Kim et al. (2023) | High | High | High | Low |

and Medicine (2021). Early detection of AD is a significant step to slow symptom progression and increase patients' life expectancy Nelson & Tabet (2015); Chu (2012). Compared to other AD biomarker modalities, EEG has unique advantages (Table 1), such as high accessibility through portable devices, high safety due to its non-invasive nature, and high affordability with low hardware and operational costs Ieracitano et al. (2019a). EEG also enables long-term monitoring of disease progression with high temporal resolution and real-time capabilities Ahmed et al. (2025). The limited detection accuracy of EEG represents a key bottleneck for its real-world deployment, which is the central challenge to address in this paper. Compared to traditional manual biomarker (e.g., phase shift Wang et al. (2017), power spectral density Fahimi et al. (2017), and Shannon entropy Azami et al. (2019)), recent deep learning methods achieved better classification performance with convolutional neural networks (CNNs) Roncero-Parra et al. (2024), graph neural networks (GNNs) Klepl et al. (2023), and Transformers Wang et al. (2024c) (Related Work; Appendix A).

However, existing deep learning methods for EEG-based AD detection still face critical limitations in both practical application and theoretical foundations. From an application perspective, **large,**

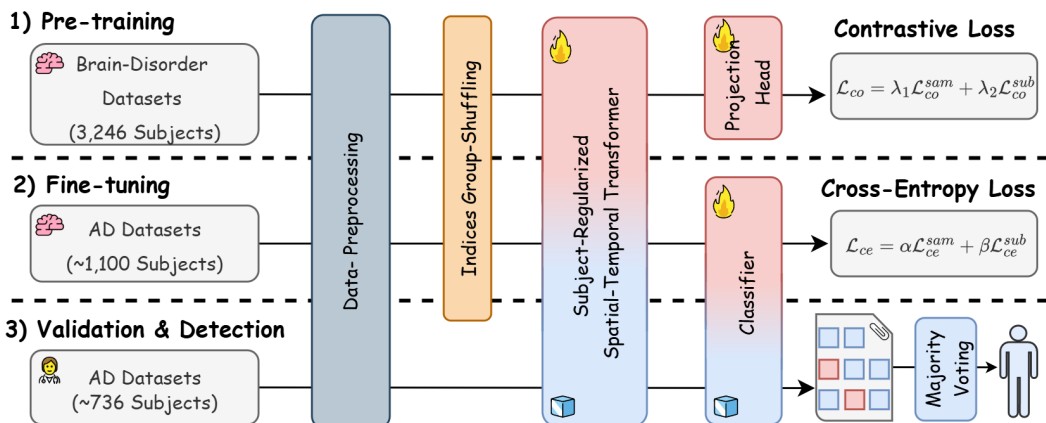

**Figure 1: LEAD overview.** We pre-train LEAD on 12 neurological disorder datasets using both sample- and subject-level contrastive losses. The model is then fine-tuned on 4 AD datasets (60% of subjects), with the remaining subjects used for validation (20%) and final detection/testing (20%). Both pre-training and fine-tuning incorporate a novel index group-shuffling technique to enhance subject-level learning. We also designed a subject-level cross-entropy loss $\mathcal{L}_{ce}^{sub}$ to encourage the model to make consistent and correct predictions for all samples belonging to the same subject.

**high-quality datasets remain scarce**. The expense and complexity of collecting EEG-AD data lead to most studies involving relatively small cohorts, dominantly ranging from 20 to 200 subjects Aviles et al. (2024). Such a limited subject scale makes it challenging to demonstrate the robustness and generalizability of models. From a theoretical perspective, few deep learning methods are **explicitly designed for subject-level** detection, which is the cornerstone of clinical detection. Most existing approaches focus on sample-level, segmenting each subject's trials into short EEG samples for feature extraction and representation learning, with performance reported at the sample-level. Some studies apply post-processing majority voting to aggregate sample-level predictions into subject-level detections Ieracitano et al. (2019a); Barbera et al. (2024). However, as highlighted by prior work Wang et al. (2024c), strong sample-level performance does not necessarily translate into accurate subject-level detection when using a non-trainable voting mechanism. Imbalances in the recording lengths of individual subjects can lead to overfitting to specific participants, ultimately limiting the model's subject-level detection capability, which is essential for real-world clinical applications.

To address the aforementioned limitations, we propose LEAD, the **first large-scale foundation model for EEG-based AD detection**. We integrated 7 highly heterogeneous EEG-AD datasets, creating a new unified dataset with a total of 2,255 subjects and 442.47 hours of recordings. To the best of our knowledge, this represents **the world's largest EEG-based AD detection dataset**, approximately **20 times larger** than the datasets commonly used in prior research. Building upon this unique resource, we design a comprehensive pipeline for subject-level AD detection. Beyond standard preprocessing steps, such as artifact removal, channel alignment, and filtering, we introduce a *multi-scale segmentation strategy* tailored for subject-level detection, enabling the model to capture long-term features across multiple temporal scales. In addition, we propose a *subject-regularized spatio-temporal transformer* trained with a novel *subject-level cross-entropy loss*, complemented by an *indices group-shuffling* algorithm to enhance subject-level learning. While our unified AD dataset is the largest of its kind, its scale might be insufficient for training a foundation model from scratch. To overcome this, we additionally curate a large-scale pre-training EEG dataset from patients with related neurological disorders, such as Parkinson's disease (PD), depression(DEP), and attention-deficit/hyperactivity disorder (ADHD). We posit that EEG signals from these neurological conditions share underlying patterns with AD, enabling LEAD to learn a robust, general-purpose representation of neurological abnormalities. Guided by neuroscientific knowledge, we employ a contrastive learning framework to leverage this domain-relevant data, creating a powerful base model that is then fine-tuned for AD detection. As shown in Figure 1, the complete pipeline of LEAD involves pre-training on 12 brain-disorder datasets using both sample- and subject-level contrastive learning, followed by fine-tuning on 60% of subjects from 4 AD datasets with joint sample- and subject-level cross-entropy losses, while the remaining 40% are reserved for validation and evaluation.

The key contributions of this work are summarized as follows: 1) **Largest EEG-based AD detection dataset.** We curate the largest dataset for EEG-based AD detection to date, far surpassing prior work.

2) **Unified subject-level AD detection pipeline.** We design a subject-level deep-learning pipeline with multi-scale segmentation, a subject-regularized spatio-temporal transformer, and AD-knowledge-guided contrastive pre-training on domain-relevant datasets. 3) **First foundation model.** Built on this unique dataset and pipeline, we develop the first foundation model for EEG-based AD detection, adaptable to diverse channel topologies, EEG devices, demographics, etc. 4) **Superior performance.** LEAD achieves state-of-the-art subject-level results on 4 AD datasets under challenging subject-independent cross-validation.

## 2 METHOD

**Problem Formulation.** EEG-based AD detection aims to determine whether a patient has AD. While most deep learning methods in this field focus on classifying small, individual segments of the EEG (sample-level classification), our approach centers on subject-level detection, which is obviously more clinically relevant Aviles et al. (2024). *(1) Sample-Level Classification:* Consider an input EEG sample $x \in \mathbb{R}^{T \times C}$, where $T$ and $C$ denote the number of timesteps and channels. The goal is to learn a model that predicts the corresponding label $y \in \mathbb{R}^K$ for each input sample. Specifically, $K$ represents the number of diagnostic categories, such as AD, Healthy Controls(HC), Mild Cognitive Impairment(MCI), or other types and stages of dementia. *(2) Subject-Level Detection:* Each sample is also associated with a subject ID $s$ indicating which subject it belongs to. For subject-level detection, we aggregate sample-level predictions via majority voting: the final label assigned to a subject is the class that appears most frequently among all of that subject's samples.

### 2.1 DATA CURATION

We categorize EEG datasets utilized in this paper into two groups: those containing data from subjects with Alzheimer's Disease (**AD** datasets) and those without (**Non-AD** datasets).

**AD Datasets.** Despite the promise of EEG for AD detection, the field remains critically limited by the scarcity of accessible datasets. To address this, we conducted a comprehensive review of relevant publications and public EEG repositories (e.g., OpenNeuro, Dryad) published before 2025. This effort identified 8 public datasets, of which we retained 5 that contained at least 30 subjects and more than 1 minute of recording per subject to ensure generalizability. We also incorporated 2 private datasets from collaborators. In total, we utilize 7 large-scale AD datasets, most of which consist of resting-state recordings: **AD-Auditory** Lahijanian et al. (2024), **BrainLat** Prado et al. (2023), **P-ADIC** Shor et al. (2021), **ADFTD** Miltiadous et al. (2023b), **CNBPM** Amezquita-Sanchez et al. (2019), **Cognision**, and **CAUEEG** Kim et al. (2023). These datasets comprise *2,255 subjects and 442.47 hours* of recordings, forming the world's largest EEG-AD corpus reported to date.

**Non-AD Datasets.** While the inclusion of AD datasets represents a major step forward, their scale is still limited for large-scale pre-training, and a portion must be reserved for downstream evaluation. To further expand our training resources, we curated 9 *domain-relevant* non-AD datasets. Unlike existing EEG foundation models that mix paradigms such as resting-state (RS), motor imagery (MI), and event-related potentials (ERP) Jiang et al. (2024b); Wang et al. (2024a), we deliberately focus on datasets that are consistent with AD detection. This decision is motivated by the fact that different paradigms reflect distinct neuroscientific processes and cortical activations, usually requiring paradigm-specific preprocessing pipelines. For example, Z-score normalization is widely used in RS EEG but can lead to information loss when applied to ERP data. Combining data from heterogeneous paradigms can introduce conflicting patterns and diminish the utility of pre-training resources. Our curated non-AD datasets contain recordings from both healthy subjects and patients with other brain disorders (e.g., PD, DEP, ADHD). Specifically, we include: **BACA-RS** Getzmann et al. (2024), **Depression** Cavanagh et al. (2019), **FEPCR** Phalen et al. (2020), **MCEF-RS** Chenot et al. (2024), **PD-RS** Singh et al. (2023), **PEARL-Neuro** Dzianok & Kublik (2024), **SRM-RS** Hatlestad-Hall et al. (2022), **TDBrain** Van Dijk et al. (2022), and **TUEP** Veloso et al. (2017). Collectively, they comprise *2,827 subjects and 632.63 hours* of EEG. Most of these datasets were acquired under resting-state or resting-state-like paradigms, ensuring consistency with our downstream AD detection.

**Unified Preprocessing.** The datasets are highly heterogeneous, with substantial variability in channel numbers, sampling rates, and recording lengths. To utilize them for training, we apply the following unified preprocessing pipeline: *(1) Artifact Removal:* Independent component analysis

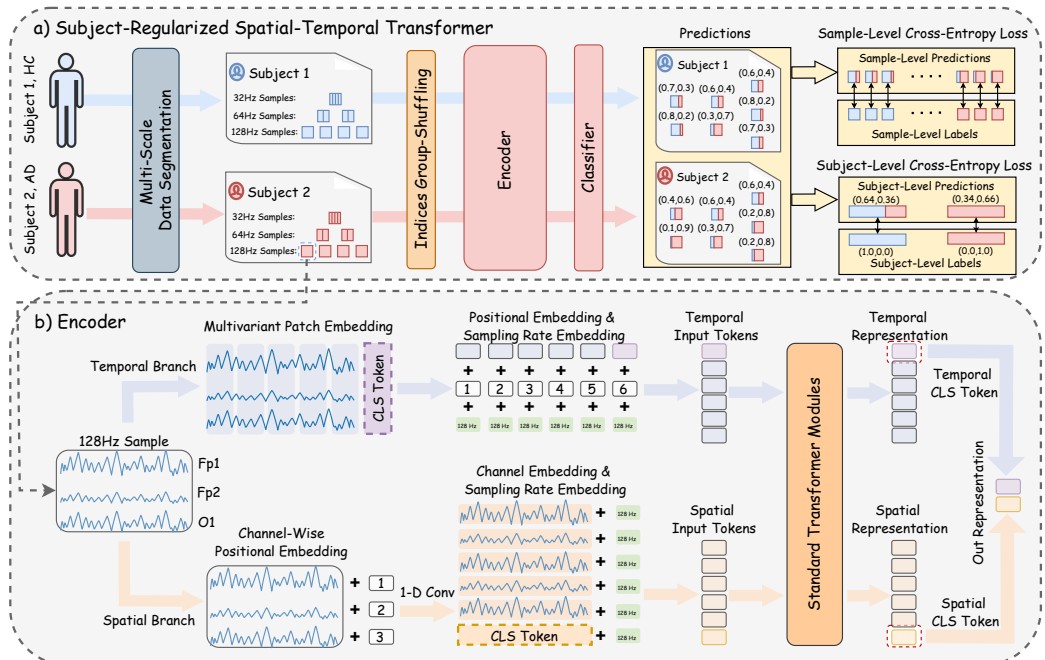

**Figure 2: a) Subject-Regularized Spatial-Temporal Transformer.** Each subject's recordings are segmented into 1s, 2s, and 4s windows (128Hz, 64Hz, 32Hz) using a multi-scale segmentation strategy. The indices group-shuffling ensures that each training batch contains multiple samples from the same subject, thereby facilitating subject-level learning. The classifier generates sample-level predictions, which are then aggregated to form subject-level predictions. **b) Encoder.** We present the encoder architecture, taking a 128Hz sample as an example. The temporal branch embeds multivariate EEG patches with a temporal `[CLS]` token, while positional and sampling-rate embeddings are added to produce temporal input tokens. The spatial branch first adds channel-wise positional embedding, then expands channels via 1D convolutions, embeds channels, and appends a spatial `[CLS]` token to the embeddings. We design a sampling rate embedding to provide the model with information regarding the segmentation scale. We concatenate the temporal and spatial `[CLS]` tokens as the final representation. d temporal and spatial representations, from which we take their `[CLS]` tokens and concatenate them as the final representation.

(ICA) combined with ICLabel Pion-Tonachini et al. (2019) is used to automatically identify and remove components linked to eye blinks, muscle activity, and cardiac signals. *(2) Channel Alignment:* All datasets are mapped to the 19-channel 10–20 montage (Fp1, Fp2, F7, F3, Fz, F4, F8, T3/T7, C3, Cz, C4, T4/T8, T5/P7, P3, Pz, P4, T6/P8, O1, O2) Homan et al. (1987) via channel interpolation or 3D coordinate mapping in MNE Gramfort et al. (2013). As most EEG datasets at least include these channels, alignment allows seamless cross-dataset training and evaluation. *(3) Band-Pass Filtering:* Each trial (i.e., recording) is band-pass filtered between 0.5 Hz and 45 Hz to remove slow drifts and high-frequency noise. *(4) Frequency Alignment:* All recordings are resampled to a uniform 128 Hz. *(5) Data Segmentation:* We newly introduce a **multi-scale segmentation** strategy. Trials at 128 Hz are further downsampled to 64 Hz and 32 Hz. Each trial is segmented into 128-timestep windows with 50% overlap, corresponding to 1-, 2-, and 4-second samples, respectively. This design not only captures multi-scale temporal dynamics but also increases per-subject sample counts, improving subject-level learning. A sampling rate marker $r$ is attached to each sample for subsequent use. *(6) Z-Score Normalization:* Finally, each channel of every segmented sample is normalized independently to zero mean and unit variance. The statistics of the processed datasets are summarized in Table 2. Additional rationale and dataset-specific preprocessing details are provided in Appendix J.

## 2.2 Subject-Regularized Spatial-Temporal Transformer

We propose a subject-regularized transformer architecture for subject-level AD detection. As shown in Figure 2, the encoder $f(\cdot)$ consists of two parallel branches: the *temporal branch*, which treats multivariant EEG patches as input tokens, and the *spatial branch*, which treats the entire time series of a channel as a single input token. Both branches employ a standard encoder-only transformer

architecture composed of self-attention layers, layer normalization, feed-forward networks, and a classifier for *sample-level predictions*. The sample-level predictions are averaged per subject to get *subject-level predictions*. Beyond the sample-level loss function, we introduce a subject-level loss together with an indices group-shuffling algorithm to enhance subject-level prediction.

**Temporal Branch.** Given an input EEG sample $\boldsymbol{x} \in \mathbb{R}^{T \times C}$ and patch length $L$, where $T$ and $C$ denote the number of timesteps and channels, respectively. We first divide the input sample into $N$ non-overlapping multivariant patches, obtaining: $\boldsymbol{x}^{\mathrm{t}} \in \mathbb{R}^{N \times (L \cdot C)}$. Zero padding is applied to ensure that $T$ is divisible by $L$, yielding $N = \lceil \frac{T}{L} \rceil$. The patches $\boldsymbol{x}^{\mathrm{t}}$ are projected into $D$-dimensional embeddings $\boldsymbol{e}^{\mathrm{t}}$ via a linear mapping $\boldsymbol{W}$. Considering a fixed positional embedding $\boldsymbol{W}_{\mathrm{pos}}^{\mathrm{t}}$, we have $\boldsymbol{e}^{\mathrm{t}} = \boldsymbol{x}^{\mathrm{t}} \boldsymbol{W} + \boldsymbol{W}_{\mathrm{pos}}^{\mathrm{t}}$, where $\boldsymbol{e}^{\mathrm{t}} \in \mathbb{R}^{N \times D}$, $\boldsymbol{W} \in \mathbb{R}^{(L \cdot C) \times D}$, and $\boldsymbol{W}_{\mathrm{pos}}^{\mathrm{t}} \in \mathbb{R}^{N \times D}$. A temporal [CLS] token $\boldsymbol{u}^{\mathrm{t}} \in \mathbb{R}^{1 \times D}$ is then initialized and appended to $\boldsymbol{e}^{\mathrm{t}}$, resulting in embeddings of shape $\boldsymbol{e}^{\mathrm{t}} \in \mathbb{R}^{(N+1) \times D}$. We then map the sampling-rate marker $r \in \mathbb{R}$ of each sample to a corresponding embedding $\boldsymbol{e}^{\mathrm{r}} \in \mathbb{R}^{1 \times D}$ and add it (via broadcasting) to all $N + 1$ tokens in $\boldsymbol{e}^{\mathrm{t}}$. This step enables the model to differentiate samples with different sampling rates produced by the multi-scale segmentation in pre-processing. The final temporal embeddings $\boldsymbol{e}^{\mathrm{t}}$ serve as the input tokens for the temporal branch self-attention. After $M$ encoding layers, we obtain the temporal branch's representations $\boldsymbol{h}^{\mathrm{t}}$.

**Spatial Branch.** Take the same EEG sample $\boldsymbol{x} \in \mathbb{R}^{T \times C}$ as in the temporal branch, we first transpose the sample and add a fixed channel-wise positional embedding $\boldsymbol{W}_{\mathrm{pos}}^{\mathrm{c}}$ to obtain $\boldsymbol{x}^{\mathrm{c}} = (\boldsymbol{x})^T + \boldsymbol{W}_{\mathrm{pos}}^{\mathrm{c}}$, where $\boldsymbol{x}^{\mathrm{c}}, \boldsymbol{W}_{\mathrm{pos}}^{\mathrm{c}} \in \mathbb{R}^{C \times T}$. Unlike the temporal branch, we add positional embedding directly to the raw transposed input data here, as subsequent operations disrupt the original channel order. For a scaled channel number $F$ and embedding dimension $D$, we first expand the channel number using a 1-D convolution $\boldsymbol{W}_1$, then map the entire channel into a latent embedding using a linear projection $\boldsymbol{W}_2$: $\boldsymbol{e}^{\mathrm{c}} = (\boldsymbol{W}_1 \boldsymbol{x}^{\mathrm{c}}) \boldsymbol{W}_2$, where $\boldsymbol{e}^{\mathrm{c}} \in \mathbb{R}^{F \times D}$, $\boldsymbol{W}_1 \in \mathbb{R}^{F \times C}$, and $\boldsymbol{W}_2 \in \mathbb{R}^{T \times D}$. Same as the temporal branch, we initialize a spatial [CLS] token $\boldsymbol{u}^{\mathrm{c}} \in \mathbb{R}^{1 \times D}$ and append it to $\boldsymbol{e}^{\mathrm{c}}$, resulting in embeddings of shape $\boldsymbol{e}^{\mathrm{c}} \in \mathbb{R}^{(F+1) \times D}$. The sampling rate embedding $\boldsymbol{e}^{\mathrm{r}} \in \mathbb{R}^{1 \times D}$ is also added (via broadcasting) to all $F + 1$ tokens in $\boldsymbol{e}^{\mathrm{c}}$. The final spatial embeddings $\boldsymbol{e}^{\mathrm{c}}$ serve as the input tokens for the spatial branch self-attention. After $M$ encoding layers, we obtain the spatial branch's representations $\boldsymbol{h}^{\mathrm{c}}$.

**Classifier.** For an input EEG sample $\boldsymbol{x}$, we obtain the temporal representation $\boldsymbol{h}^t$ and the spatial representation $\boldsymbol{h}^c$ from the corresponding branches. We concatenate the the temporal and spatial [CLS] token of them to form the final representation $\boldsymbol{h} = [\boldsymbol{h}^t[-1] \,\|\, \boldsymbol{h}^c[-1]]$ of the backbone encoder $f(\cdot)$, where $[\cdot \,\|\, \cdot]$ denotes concatenation and $\boldsymbol{h} \in \mathbb{R}^{2D}$. The resulting representation $\boldsymbol{h}$ is passed through a MLP classifier $c(\cdot)$ to get sample-level predictions $\hat{\boldsymbol{y}}$ and compute the *sample-level cross-entropy loss* $\mathcal{L}_{\mathrm{ce}}^{\mathrm{sam}} = \mathrm{CrossEntropy}(\hat{\boldsymbol{y}}, \boldsymbol{y})$, where $\boldsymbol{y}$ is the ground-truth label of each sample.

**Subject-Level Cross-Entropy Loss.** Previous studies have shown that high sample-level prediction accuracy does not necessarily mean high subject-level detection performance Wang et al. (2024c), which may stem from imbalances in the number of samples per subject and inherent subject-specific variability. To address this issue, we newly design a subject-level cross-entropy loss to encourage samples from the same subject to have consistent predictions. Each sample $\boldsymbol{x}$ is associated with a subject identifier $s$, indicating the subject to which it belongs. Within a batch, we first identify the unique subjects, then iteratively compute each sub-

---

**Algorithm 1** Subject-level Cross-Entropy Loss

**Require:** Predictions $\hat{\boldsymbol{y}} \in \mathbb{R}^{B \times K}$, labels $\boldsymbol{y} \in \mathbb{R}^{B \times K}$, subject IDs $\boldsymbol{s} \in \mathbb{R}^{B}$
**Ensure:** Subject-level loss $\mathcal{L}_{\mathrm{ce}}^{\mathrm{sub}}$

1: $\mathcal{U} \leftarrow \mathrm{Unique}(\boldsymbol{s})$    ▷ set of subject IDs in the batch
2: $\mathcal{Z} \leftarrow [\,], \ \mathcal{Y} \leftarrow [\,]$    ▷ lists for mean predictions/labels
3: **for all** $u \in \mathcal{U}$ **do**
4:     $\mathcal{M} \leftarrow \{\, i \mid \boldsymbol{s}_i = u \,\}$    ▷ subject $u$ sample indices
5:     $\bar{\boldsymbol{z}} \leftarrow \frac{1}{|\mathcal{M}|} \sum_{i \in \mathcal{M}} \hat{\boldsymbol{y}}_i$    ▷ subject $u$ mean prediction
6:     $\boldsymbol{y}_u \leftarrow \boldsymbol{y}_{i^*}, \ i^* \in \mathcal{M}$    ▷ subject $u$ label
7:     Append $\bar{\boldsymbol{z}}$ to $\mathcal{Z}$ and $\boldsymbol{y}_u$ to $\mathcal{Y}$
8: $\mathcal{L}_{\mathrm{ce}}^{\mathrm{sub}} \leftarrow \mathrm{CrossEntropy}(\mathcal{Z}, \mathcal{Y})$
9: **return** $\mathcal{L}_{\mathrm{ce}}^{\mathrm{sub}}$

---

ject's prediction by averaging the model's predictions across all samples sharing the same subject ID $s$. The *subject-level cross-entropy loss* $\mathcal{L}_{\mathrm{ce}}^{\mathrm{sub}}$ is then calculated as the cross-entropy between these subject-level predictions and their corresponding subject labels (equal to any sample with ID $s$, as discussed in Appendix B). The pseudocode for computing this loss is provided in Algorithm 1.

**Indices Group-Shuffling.** In practical training with a large number of subjects, the probability that multiple samples from the same subject appear within the batch decreases, reducing the effectiveness of subject-level learning. To mitigate this issue, we newly introduce an indices group-shuffling

algorithm that dynamically reorders the indices of each epoch. The algorithm ensures that, for every subject included in a batch, at least a predefined group size of samples from that subject is present, while maintaining enough randomness. The pseudocode is provided in Algorithm 2.

**Overall Loss Function.** The total subject-regularized cross-entropy loss for supervised learning or fine-tuning is $\mathcal{L}_{ce} = \alpha\mathcal{L}_{ce}^{sam} + \beta\mathcal{L}_{ce}^{sub}$, where $\alpha, \beta \in [0, 1]$ control the weight of two losses.

**Majority Voting.** Since our model is trained and makes initial predictions on segmented, fixed-length EEG samples, a post-processing step is generally required during the testing stage to convert these sample-level classifications into a final subject-level detection. A commonly used method is the majority vote mechanism Ieracitano et al. (2019a); Barbera et al. (2024). Specifically, for all samples from the same subject, we identify the majority label and assign it to that subject.

---

**Algorithm 2** Indices Group-Shuffling

**Require:** Subject IDs $s \in \mathbb{R}^N$, group size $g$, batch size $b$
**Ensure:** A shuffled index permutation $\pi \in \{1, \ldots, N\}^N$

1: $p \leftarrow \text{argsort}(s)$ ▷ sort indices by subject IDs
2: $\mathcal{G} \leftarrow [\,]$ ▷ list of groups
3: **for** $i \leftarrow 1$ **to** $N$ **step** $g$ **do**
4: $\quad \mathcal{G} \leftarrow \mathcal{G} \cup \{\, p[\, i : \min(i + g - 1, N)\,]\,\}$ ▷ groups
5: $\text{Shuffle}(\mathcal{G})$ ▷ shuffle the order of groups
6: $q \leftarrow \text{Concat}(\mathcal{G})$ ▷ flatten groups
7: $\mathcal{B} \leftarrow [\,]$ ▷ list of batches
8: **for** $j \leftarrow 1$ **to** $N$ **step** $b$ **do**
9: $\quad u \leftarrow q[\, j : \min(j + b - 1, N)\,]$ ▷ form a batch
10: $\quad \text{Shuffle}(u)$ ▷ shuffle indices within the batch
11: $\quad \mathcal{B} \leftarrow \mathcal{B} \cup \{u\}$
12: $\pi \leftarrow \text{Concat}(\mathcal{B})$ ▷ flatten batches back
13: **return** $\pi$

---

### 2.3 AD-Knowledge Guided Self-Supervised Contrastive Pretraining

We employ both **sample-level** and **subject-level** medical contrastive learning for self-supervised pre-training Wang et al. (2023). The sample-level contrasting performs instance discrimination, treating different augmented views of the same sample as positive pairs and views from other samples as negative pairs. The augmentation methods are described in Appendix E. The subject-level contrasting leverages subject IDs as guidance: samples from the same subject are treated as positive pairs, while those from different subjects are treated as negative pairs. Subject-level contrastive learning is particularly well-suited to our goal of capturing subject-level representations for AD detection, under the assumption that all samples from a subject share the same label (See preliminaries in Appendix B). We again incorporate the indices group-shuffling algorithm (Sec. 2.2) to increase the probability that samples with the same subject ID co-occur within a mini-batch. The overall self-supervised pre-training objective is a weighted sum of the two contrastive losses: $\mathcal{L}_{co} = \lambda_1\mathcal{L}_{co}^{sam} + \lambda_2\mathcal{L}_{co}^{sub}$, where $\lambda_1 + \lambda_2 = 1$, and $\lambda_1, \lambda_2 \in [0, 1]$ control the relative importance. Details of the contrastive loss formulations are provided in Appendix D.

## 3 Experiments

### 3.1 Setup

**Data Preprocessing.** The preprocessing pipeline for each dataset is presented in section 2.1, with a more detailed dataset and preprocessing description available in Appendix J. The statistics for the processed datasets are summarized in Table 2. For pre-training, we leverage all 9 Non-AD datasets together with 3 AD datasets, resulting in a total of *3,246 subjects, 730.48 hours, and 9,185,969 128-timestamp samples*. The remaining 4 AD datasets are reserved for downstream evaluation, resulting in a total of *1,836 subjects, 344.62 hours, and 4,246,256 128-timestamp samples*.

**Baselines.** We compare our method against 10 baselines, including 1 manual feature-extraction approach, 7 supervised deep-learning models, and 2 large EEG foundation models. These selected baselines are state-of-the-art methods or have shown strong performance in EEG classification tasks. The feature-based method extracts **Statistical**, **Spectral**, **Power**, and **Complexity** features (see Appendix I) commonly used for EEG-based AD detection, followed by classification with a linear classifier. The 7 supervised learning methods include **EEGNet** Lawhern et al. (2018), **TST** Zerveas et al. (2021), **EEGInception** Zhang et al. (2021), **EEGConformer** Song et al. (2022), **BIOT** Yang et al. (2024), **Medformer** Wang et al. (2024b), and **MedGNN** Fan et al. (2025). The 2 large EEG foundation models are **LaBraM** Jiang et al. (2024b) and **CBraMod** Wang et al. (2025).

**Table 2: Dataset Statistics.** All datasets are downsampling to 128Hz (200 Hz for LaBraM and CBraMod). Additional downsampling to 64Hz and 32Hz is applied if using multi-scale segmentation. Each sample contains 128 timesteps after segmentation, with all signals aligned to 19 channels. Abbreviations: **ASSR**: Auditory Steady-State Response; **RS**: Resting State; **HV**: Hyperventilation; **PS**: Photic Stimulation. **HC** - Healthy Controls; **AD**: Alzheimer's Disease; **FTD**: Frontotemporal Dementia; **PD**: Parkinson's disease; **MS**: Multiple Sclerosis; **MCI**: Mild Cognitive Impairment; **SCZ**: Schizophrenia; **DEP**: Depression; **DEM**: Dementia; **FEP**: First Episode Psychosis; **MDD**: Major Depressive Disorder; **ADHD**: Attention Deficit Hyperactivity Disorder.

| Category | Datasets | Paradigm | #Subjects | Sampling Rate | #Channels | Total Time | #Samples | Classes | Tasks |
|---|---|---|---|---|---|---|---|---|---|
| | AD-Auditory | ASSR | 35 | 250->128 Hz | 19 | 5.20h | 65,450 | HC, AD, MCI, | Pre-train |
| | BrainLat | RS | 135 | 512->128 Hz | 19 | 17.26h | 216,976 | HC, AD, FTD, PD, MS | Pre-train |
| | P-ADIC | RS | 249 | 500->128 Hz | 19 | 75.39h | 948,961 | HC, AD, MCI, SCZ, DEP | Pre-train |
| AD | ADFTD | RS, PS | 88 | 500->128 Hz | 19 | 26.64h | 335,332 | HC, AD, FTD | Fine-tune |
| | CNBPM | RS | 189 | 256->128 Hz | 19 | 19.51h | 244,926 | HC, AD, MCI | Fine-tune |
| | Cognision | RS | 180 | 125->128 Hz | 7->19 | 9.01h | 112,860 | HC, AD | Fine-tune |
| | CAUEEG | RS, HV, PS | 1379 | 200->128 Hz | 19 | 289.46h | 3,553,138 | HC, DEM, MCI, etc. | Fine-tune |
| | BACA-RS | RS | 608 | 1000->128 Hz | 65->19 | 169.80h | 2,137,450 | HC | Pre-train |
| | Depression | RS | 122 | 500->128 Hz | 66->19 | 6.69h | 83,708 | HC, DEP | Pre-train |
| | FEPCR | RS | 143 | 1000->128Hz | 64->19 | 12.27h | 154,062 | HC, FEP | Pre-train |
| | MCEF-RS | RS | 165 | 512->128Hz | 64->19 | 14.14h | 177,484 | HC | Pre-train |
| Non-AD | PD-RS | RS | 149 | 500->128 Hz | 60->19 | 6.64h | 83,017 | HC, PD | Pre-train |
| | PEARL-Neuro | RS | 79 | 1000->128 Hz | 127->19 | 14.36h | 180,642 | HC | Pre-train |
| | SRM-RS | RS | 109 | 1024->128 Hz | 64->19 | 9.10h | 114,333 | HC | Pre-train |
| | TDBrain | RS | 1273 | 500->128 Hz | 33->19 | 89.71h | 1,125,784 | MDD, ADHD, PD, etc. | Pre-train |
| | TUEP | RS | 179 | 256->128 Hz | 19 | 309.92h | 3,898,102 | HC, Epilepsy | Pre-train |

**Training & Parameter Settings.** Self-supervised pre-training is conducted for 50 epochs without early stopping, followed by fine-tuning for up to 200 epochs with early stopping (patience=15) based on the best sample-level F1. Batch sizes are 2048 for pre-training and 512 for supervised learning. We use AdamW with learning rates of $2 \times 10^{-4}$ (pre-training) and $1 \times 10^{-4}$ (fine-tuning), scheduled by CosineAnnealingLR. Model performance is evaluated using **sample-level and subject-level sensitivity and F1**, with subject-level metrics computed via majority voting (Section 2.2). Both sample-level and subject-level cross-entropy losses are weighted by $\alpha = \beta = 0.5$, while contrastive loss coefficients are set to $\lambda_1 = 0.25$ and $\lambda_2 = 0.75$. The cross-channel patch length is $L = 4$ and the scaled channel number is $F = 76$. For self-supervised pre-training, all subjects from the pre-training datasets are used. For supervised learning and fine-tuning, we adopt Monte Carlo cross-validation Xu & Liang (2001) with a subject-independent Wang et al. (2024b) 6:2:2 train/validation/test split, ensuring no subject overlap while preserving randomness across seeds. To enable cross-domain transfer, no subjects or datasets from pre-training are included in fine-tuning. For foundation model baselines (LaBraM and CBraMod), we fine-tune their released checkpoints to assess the benefit of domain-relevant pre-training; their downstream data are resampled to 200 Hz for consistency, while ours and other baselines use 128 Hz. Each evaluation is repeated with five random seeds (41-45), reporting the mean and standard deviation. Experiments are run on 4 NVIDIA RTX A5000 GPUs with Python 3.10 and PyTorch 2.5.1+cu121. Further details are provided in Appendix I.

## 3.2 RESULTS

**Performance Comparison with Baselines.** We compare LEAD against 10 baselines on two tasks.

*Multi-Class Classification.* We begin with a multi-class classification task to evaluate our model's ability to distinguish among dementia-related classes, including different dementia types such as FTD and AD, as well as prodromal stages such as MCI. Notably, in the CAUEEG dataset, AD is encompassed in the broader Dementia (DEM) category, which primarily consists of AD cases, along with some Vascular Dementia (VD) and very mild Parkinson's Disease Dementia (PDD). We adhere to this default class split. The experimental results are summarized in Table 3. Our method achieves the highest scores in 15 out of 16 metrics. In particular, our method consistently outperforms all other approaches across subject-level metrics, including sensitivity and F1 score. By focusing on domain-relevant datasets, our pre-training resources (approximately 730.48 hours of data) outperform LaBraM and CBraMod, despite their substantially larger pre-training corpora of roughly 2,000 hours and over 9,000 hours of EEG, respectively. Our leading performance highlights the importance of selecting suitable pre-training datasets for AD detection. *AD/DEM vs HC Classification.* We next conduct a binary classification task to evaluate the model's ability to distinguish subjects with AD or DEM from HC. The results are presented in Appendix F.1.

**EEG Paradigms Comparison.** We evaluate the ADFTD dataset across three paradigms: resting-state, photic-stimulation, and in combination. Results are shown in Table 4 and Figure 3. The combined setting substantially outperforms the individual paradigms, achieving a subject-level F1 of 75.85%

Table 3: **Multi-Class Classification Results.** Detecting different dementia types and stages: HC, AD, and FTD for ADFTD; HC, AD, and MCI for CNBPM; HC and AD for Cognision, and HC, DEM, and MCI for CAUEEG.

| Datasets | ADFTD (335,332 Samples) (88 Subjects, 3 Classes) | | CNBPM (244,926 Samples) (189 Subjects, 3 Classes) | | Cognision (112,860 Samples) (180 Subjects, 2 Classes) | | CAUEEG (3,553,138 Samples) (1187 Subjects, 3 Classes) | |
|---|---|---|---|---|---|---|---|---|
| Metrics
Methods | Sensitivity | F1 Score | Sensitivity | F1 Score | Sensitivity | F1 Score | Sensitivity | F1 Score |
| **Sample-Level Classification** | | | | | | | | |
| ManualFeature | 45.30±2.14 | 45.02±2.06 | 48.26±5.90 | 47.83±5.53 | 53.54±1.09 | 53.05±0.99 | 43.44±2.17 | 42.60±2.25 |
| EEGNet | 44.78±2.62 | 42.36±2.72 | 49.44±4.61 | 47.03±4.80 | 52.70±2.39 | 50.70±3.17 | 45.53±0.75 | 45.26±0.69 |
| TST | 54.68±3.87 | 54.33±4.51 | 61.03±5.02 | 60.45±5.44 | 56.34±2.37 | 55.80±2.75 | 49.17±1.60 | 48.90±1.56 |
| EEGInception | 62.26±2.32 | 61.61±2.03 | 60.77±2.61 | 59.56±3.67 | 60.09±1.46 | 59.98±1.34 | 50.98±1.57 | 49.85±1.88 |
| EEGConformer | 63.53±3.90 | 63.58±4.35 | 62.73±4.47 | 61.97±4.90 | 58.69±3.05 | 58.39±3.04 | 51.14±1.35 | 51.21±1.55 |
| BIOT | 58.48±2.44 | 58.28±2.52 | 54.24±5.54 | 53.77±5.29 | 54.37±2.94 | 54.15±2.95 | 48.59±1.37 | 48.22±1.62 |
| MedGNN | 60.33±3.33 | 59.98±3.37 | 63.70±2.97 | 63.09±3.31 | 60.02±2.48 | 59.96±2.50 | 49.67±1.40 | 49.63±1.46 |
| Medformer | 57.71±4.69 | 57.54±5.34 | 63.04±3.69 | 62.30±4.21 | 57.16±2.38 | 56.88±2.21 | 49.40±1.19 | 48.80±1.14 |
| LaBraM | 64.96±4.43 | 64.83±4.54 | 60.96±5.77 | 60.81±6.02 | 61.70±2.95 | 61.40±3.06 | **51.87±1.44** | **51.73±1.45** |
| CBraMod | 56.67±1.53 | 56.60±1.79 | 57.59±6.08 | 57.59±6.30 | 55.26±3.02 | 54.81±3.52 | 46.84±1.46 | 46.88±1.48 |
| **LEAD** | **67.63±3.55** | **67.41±3.77** | **64.52±3.44** | **63.89±3.70** | **62.02±2.15** | **61.87±2.24** | 51.49±1.49 | 51.22±1.57 |
| **Subject-Level Detection** | | | | | | | | |
| ManualFeature | 57.00±3.97 | 53.35±6.69 | 55.90±6.96 | 51.75±7.74 | 53.51±4.32 | 50.44±4.18 | 48.78±4.72 | 45.62±6.27 |
| EEGNet | 55.44±4.97 | 49.42±7.42 | 56.41±1.62 | 52.70±3.97 | 54.59±3.15 | 49.49±6.18 | 49.19±0.53 | 48.66±0.85 |
| TST | 69.67±5.18 | 68.16±7.32 | 58.46±9.37 | 58.39±9.10 | 62.16±7.64 | 60.89±7.98 | 57.53±2.24 | 57.08±2.64 |
| EEGInception | 73.11±4.15 | 71.14±4.93 | 60.51±6.80 | 60.09±6.57 | 63.15±5.33 | 62.95±5.37 | 58.64±2.25 | 56.49±3.51 |
| EEGConformer | 74.39±7.34 | 72.77±9.08 | 63.59±2.99 | 63.58±3.03 | 64.44±5.32 | 63.45±5.34 | 58.23±2.02 | 58.54±2.17 |
| BIOT | 72.67±6.42 | 72.43±6.77 | 60.00±3.48 | 57.17±5.92 | 56.50±4.99 | 53.89±6.81 | 55.78±3.15 | 55.28±3.66 |
| MedGNN | 67.56±5.27 | 66.06±6.69 | 64.62±2.99 | 64.31±3.21 | 64.09±5.04 | 63.84±5.16 | 55.88±1.50 | 55.99±1.64 |
| Medformer | 71.11±6.27 | 69.72±8.03 | 62.05±6.96 | 62.00±6.94 | 55.74±3.14 | 54.20±4.28 | 57.25±1.08 | 56.45±1.50 |
| LaBraM | 74.39±7.92 | 73.65±8.92 | 60.51±6.20 | 59.88±7.11 | 67.29±4.38 | 66.09±4.76 | 58.66±1.68 | 58.92±1.96 |
| CBraMod | 73.33±4.60 | 73.31±5.01 | 56.92±11.05 | 55.60±11.63 | 56.76±6.49 | 55.77±7.53 | 56.64±2.75 | 56.87±2.98 |
| **LEAD** | **76.72±5.94** | **75.77±5.81** | **64.62±5.23** | **64.34±5.11** | **70.62±4.52** | **70.30±4.45** | **59.65±1.53** | **59.61±1.82** |

versus 52.42% (resting-state) and 48.33% (photic-stimulation). While it remains unclear whether this gain arises from longer recordings or complementary paradigm features, the evidence indicates that *combining these two paradigms is non-detrimental and likely to be beneficial for AD detection.*

Table 4: **EEG Paradigms Comparison.** Each ADFTD subject was recorded under two paradigms with varying recording lengths. We evaluate model performance on each paradigm separately and in combination.

| Paradigms | Sample-Level Sensitivity | Sample-Level F1 Score | Subject-Level Sensitivity | Subject-Level F1 Score |
|---|---|---|---|---|
| Resting-State Eyes-Closed (19.40h) | 50.28±3.02 | 48.93±3.32 | 54.83±5.67 | 52.42±7.61 |
| Photic-Stimulation Eyes-Open (7.25h) | 47.95±8.18 | 47.00±8.22 | 50.83±10.08 | 48.33±10.55 |
| Combination (26.64h) = LEAD | **67.63±3.55** | **67.41±3.77** | **76.72±5.94** | **75.77±5.81** |

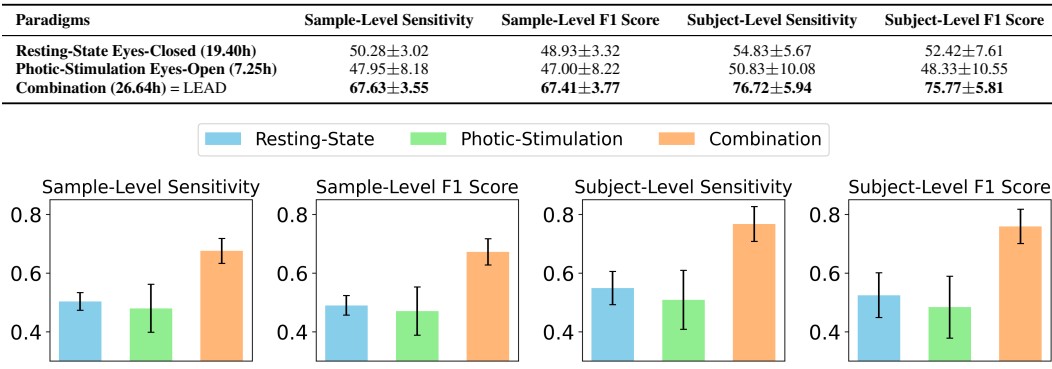

**Figure 3:** Performance comparison among different EEG paradigms on the ADFTD dataset.

**Ablation Study of Modules.** We conduct an ablation study on the ADFTD dataset to assess the contribution of each module, reporting both sample- and subject-level F1 scores (Table 5). Starting from a vanilla spatio-temporal transformer trained in a supervised manner on non-overlapping samples, we progressively add: (1) half-overlapping segmentation, (2) sample- and

Table 5: **Ablation Study of Modules.** We gradually add each component to see the performance change on ADFTD datasets of both sample-level and subject-level F1 scores.

| Models | Sample-Level F1 Score | Subject-Level F1 Score |
|---|---|---|
| Vanilla Backbone | 57.95±4.27 | 73.11±6.89 |
| + Half-Overlapping | 58.35±3.42 ↑0.40 | 72.12±8.69 ↓0.99 |
| + Sample-Level Pre-training | 61.52±4.51 ↑3.17 | 72.42±10.03 ↑0.30 |
| + Subject-Level Pre-training | 64.94±4.09 ↑3.42 | 75.49±6.09 ↑3.07 |
| + Multi-Scale Segmentation | 67.87±4.20 ↑2.93 | 74.96±6.18 ↓0.53 |
| + Subject Cross-Entropy Loss | 66.24±4.67 ↓1.63 | 75.30±5.79 ↑0.34 |
| + Indices Group-Shuffling = LEAD | **67.41±3.77** ↑1.17 | **75.77±5.81** ↑0.47 |

subject-level contrastive pre-training, (3) multi-scale segmentation, (4) subject-level cross-entropy loss, and (5) indices group-shuffling. Most modules yield consistent gains in sample-level F1, with the largest improvements from contrastive pre-training (up to +3.42%) and multi-scale segmentation (+2.93%). Subject-level F1 shows strong variability, echoing prior findings Wang et al. (2024c)

that sample-level gains may not always transfer to subject-level. Notably, the subject-level loss and group-shuffling strategy alleviate this mismatch, improving subject-level performance while introducing only a modest trade-off at the sample-level.

**Ablation Study of Pre-training Datasets.** We also conduct an ablation study to evaluate the effectiveness of each domain-relevant pre-training dataset. The results are presented in Appendix G.1

**Leave-One-Subject-Out Analysis.** We further explore subject-specific performance across demographics, recording length, MMSE score, and clinical subgroups under the LOSO setting on ADFTD. The full results of each subject are in Appendix F.2 and the averaged subject-level accuracy of all subjects is 90.91%. A comparison among different groups is in Table 6 and Figure 4. Female subjects show higher accuracy than males, while age has little effect. Longer recordings (>0.32h) substantially improve detection performance, highlighting the importance of recording duration. Mid-range MMSE scores (18–24) are harder to identify, suggesting less distinctive neural patterns in this group. Across diagnostic classes, HC subjects are most reliably identified, whereas FTD subjects remain the most challenging to distinguish, reflecting the inherent difficulty in distinguishing FTD from AD and HC.

**Table 6: LOSO Results Comparison.** We report results in different subject groups in terms of demographics, recording length, MMSE score, and dementia category.

| Label | Gender | | Age | | Length | | MMSE | | | Dementia Class | | |
|---|---|---|---|---|---|---|---|---|---|---|---|---|
| Group | M | F | <65 | ≥65 | <0.32h | >=0.32h | 0-17 | 18-24 | 25-30 | AD | HC | FTD |
| Subject Number | 44 | 44 | 38 | 50 | 45 | 43 | 13 | 42 | 33 | 36 | 29 | 23 |
| Sample-level Accuracy | 78.59% | **83.08%** | 80.04% | **81.44%** | 76.55% | **85.32%** | 82.91% | 77.76% | **83.93%** | 80.72% | **90.14%** | 69.29% |
| Subject-level Accuracy | 88.64% | **93.18%** | **92.11%** | 90.00% | 84.44% | **97.67%** | **92.31%** | 90.48% | 90.91% | 91.67% | **96.55%** | 82.61% |

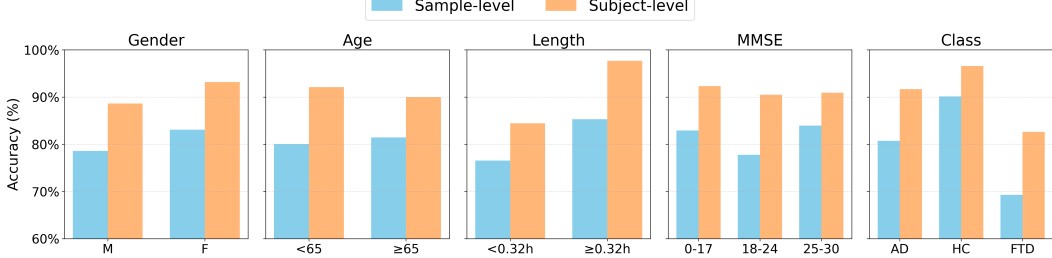

**Figure 4:** Performance comparison among different groups on the ADFTD dataset using the LOSO setup.

**Channel Importance Analysis.** We select the top-performing subjects (Sub-027, Sub-064, Sub-067) under the LOSO setting for each clinical group and plot their channel importance in Figure 5. For AD and HC subjects, the patterns are less interpretable, though AD importance may relate to affected brain regions. In contrast, the FTD subject shows a clear focus on frontotemporal electrodes.

## 4 CONCLUSION AND LIMITATIONS

This paper introduces LEAD, the first foundation model for EEG-based AD detection, trained on the world's largest EEG-AD corpus. Our methodology integrates a full AD detection pipeline, including a multi-scale segmentation strategy, a subject-regularized spatio–temporal transformer with both sample- and subject-level cross-entropy losses, an

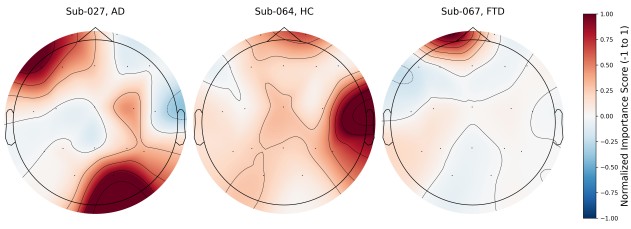

**Figure 5:** Channel importance of 3 top-performing subjects.

indices group-shuffling algorithm, and pre-training on domain-relevant brain-disorder datasets. A final majority-voting mechanism is employed for subject-level detection. Our method achieves state-of-the-art subject-level performance across all four AD datasets, surpassing existing foundation models despite their use of significantly larger pre-training resources. Furthermore, our analyses highlight that combining resting-state and photic-stimulation paradigms can be beneficial, and that longer per-subject recordings are critical for improved performance. Additional discussion of these findings, as well as the limitations and future research directions, is provided in Appendix H.

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

# APPENDIX A    RELATED WORK

## A.1    EEG-BASED AD DETECTION

In the last two decades, EEG-based AD detection has followed two main research directions: manual biomarker extraction and deep learning representation.

**Manual Biomarker Extraction:** This research direction aims to identify potential biomarkers in EEG signals of AD patients and use simple classifiers, such as Multi-Layer Perceptrons (MLP) and Support Vector Machines (SVM), to differentiate these features from normal healthy subjects. Different types of EEG features are used, including statistical features like Mean, Skewness, Kurtosis, and Standard Deviation Tzimourta et al. (2019b;a); Kulkarni & Bairagi (2017); Kanda et al. (2014); Waser et al. (2013); Tylova et al. (2013); Mora-Sánchez et al. (2019), spectral features like Phase Shift, Phase Coherence, Bispectrum, and Bicoherence Wang et al. (2017); Cassani et al. (2014); Wang et al. (2015); Fraga et al. (2013); Tait et al. (2019); Waser et al. (2016); Trambaiolli et al. (2011), power features like Power Spectrum Density, Relative Band Power, Ratio of EEG Rhythm, and Energy Fahimi et al. (2017); Schmidt et al. (2013); Liu et al. (2016); Kanda et al. (2014), as well as complexity features like Shannon Entropy, Tsallis Entropy, and Permutation Entropy Garn et al. (2015); Azami et al. (2019); Tylová et al. (2018); Coronel et al. (2017); Al-Nuaimi et al. (2018). The main advantage of this approach is its interpretability, but it suffers from limited performance.

**Deep Learning:** Compared to manual biomarker extraction, deep learning offers an alternative approach by automatically extracting useful representations for AD detection. Models such as Convolutional Neural Networks (CNNs) Li et al. (2022); Cura et al. (2022), Graph Neural Networks (GNNs) Shan et al. (2022); Klepl et al. (2023), and Transformers Wang et al. (2024c) are widely used for representation learning. Some researchers still perform manual feature extraction or transform the data before applying deep learning models. For example, the method in Ieracitano et al. (2019a) converts 5-second EEG intervals into Power Spectral Density (PSD) images and uses 2D convolutional layers on the images for feature extraction. DICE-net Miltiadous et al. (2023a) extracts relative band power and spectral coherence connectivity across five frequency bands and applies convolutional layers followed by transformers. In contrast, some studies apply deep learning methods directly to EEG data. For instance, the method in Gallego-Viñarás et al. (2024) uses semi-supervised spatiotemporal representation learning with deep learning models for AD detection based on different sleep-stage EEG data. STEADYNet Kachare et al. (2024) designs low-complexity convolutional models for AD and dementia detection, focusing on fast inference times. Research in Watanabe et al. (2024) using MNet that applies convolutional networks for feature extraction and concatenates with the relative power spectrum for AD and other dementia detection. ADformer Wang et al. (2024c) uses a multi-granularity transformer for AD detection and widely tests on 4 AD datasets.

## A.2    SELF-SUPERVISED LEARNING AND FOUNDATION MODEL IN EEG

Two main strategies are widely used for self-supervised representation learning and foundation model training in EEG: contrastive learning and mask-reconstruction.

**Contrastive Learning:** BENDR Kostas et al. (2021) follows a similar contrastive learning pipeline as Wav2Vec Baevski et al. (2020), but it is trained on EEG data. EEG2Vec Zhu et al. (2023) explores both contrastive learning and mask-reconstruction for self-supervised pre-training on EEG data. BIOT Yang et al. (2024) designs a transformer architecture for biomedical signal embedding and applies a self-supervised contrastive framework similar to BYOL Grill et al. (2020). COMET Wang et al. (2023) utilizes various data levels in biomedical time series to define positive and negative pairs.

**Mask-Reconstruction:** Neuro-BERT Wu et al. (2024) employs masked autoencoding to predict missing amplitude and phase of EEG signals during pre-training. EEG2Rep Mohammadi Foumani et al. (2024) combines a context encoder with a momentum target encoder to reconstruct context-level representations rather than raw data in self-supervised pre-training. LaBraM Jiang et al. (2024b), the first large foundation model in the EEG domain, uses a neural tokenizer to reconstruct the Fourier spectrum during self-supervised pre-training. EEGPT Wang et al. (2024a) is a foundation model for EEG representation learning that integrates reconstruction loss with an alignment loss between the encoder and momentum encoder. CBraMod Wang et al. (2025) designs a criss-cross transformer to leverage both spatial and temporal features of EEG. LUNA Döner et al. introduces a cross-attention mechanism for a topology-agnostic foundation model capable of handling heterogeneous channels.

**Other Strategies.** Recent work has begun exploring autoregressive pre-training for EEG, such as a different study also named EEGPT Yue et al. (2024). Research bridging language and EEG has also emerged, for instance, NeuroLM Jiang et al. (2024a), which treats EEG as a foreign language for representation learning.

**Application-Oriented Approaches.** Several studies focus on application-specific foundation models. Brant Zhang et al. (2023) trains a model tailored to intracranial neural signals; PPi Yuan et al. (2023) develops a self-supervised framework for subject-independent seizure detection; MIRepNet Liu et al. (2025) designs a foundation model for motor imagery classification.

## APPENDIX B  BRAIN DISEASES DETECTION WITH EEG

Neurological disease detection using EEG typically involves assigning a single label to each subject, reflecting their physiological brain state at the time of data acquisition, such as the presence or absence of AD (Conditions like seizures involve episodic labeling are not considered here). This assumes that a subject's state remains stable during the relatively short recording period(e.g, several days), meaning all data from that subject shares the same label. Exceptions arise with longitudinal datasets that track disease progression (e.g., Healthy → Mild Cognitive Impairment (MCI) → AD Vecchio et al. (2018); Ge et al. (2025)); in such cases, trials/sessions reflecting a change in diagnosis for the same subject are treated as different subjects during training. Besides, comorbid neurological conditions are often reported in brain disorders such as dementia. For instance, a patient may present with both AD and Vascular Dementia (VD) Kim et al. (2023). This introduces an inherently multi-label scenario, in which each brain condition should be predicted independently via a dedicated projection head formulated as a binary classification task.

## APPENDIX C  PERFORMANCE METRICS SELECTION

**Sensitivity** (also called **Recall** in statistics and machine learning domains) and **Specificity** are two fundamental metrics for medical diagnostic tests. Given the counts of: **TP** (true positives), **TN** (true negatives), **FN** (false negatives), **FP** (false positives), they are defined as

$$\text{Sensitivity (Recall)} = \frac{TP}{TP + FN}, \qquad \text{Specificity} = \frac{TN}{TN + FP}. \tag{1}$$

Sensitivity quantifies the ability of a model to correctly identify subjects who truly have the disease, whereas specificity measures how well it avoids false alarms among healthy subjects. For AD detection, missing a subject who may have AD could delay intervention and lead to poor clinical outcomes. Hence, we prioritize **Sensitivity** as the primary metric: it maximizes the chance of identifying every potential AD case, even at the cost of accepting more false positives.

On the other hand, the **F1 score** is a widely adopted evaluation metric in time-series classification, especially under class imbalance. It is defined as the harmonic mean of precision and recall:

$$\text{F1} = 2 \cdot \frac{\text{Precision} \cdot \text{Recall}}{\text{Precision} + \text{Recall}}, \qquad \text{Precision} = \frac{TP}{TP + FP}. \tag{2}$$

The F1 score balances the trade-off between high detection rate and the risk of false alarms, offering a more holistic view of a model's learning performance. By reporting both Sensitivity and F1 Score, we capture complementary aspects of our AD detection models: the ability to avoid missed detections and the overall balance between detecting positive cases and controlling false positives.

## APPENDIX D  CONTRASTIVE LOSS FUNCTIONS

EEG, as a type of medical time series, contains more hierarchical structures than general time series data, such as session and subject ID information. Effective pre-training strategies can leverage this information to guide the design of EEG-specific representation learning frameworks under appropriate assumptions. In this work, we utilize two contrastive modules, **sample-level** and **subject-level** contrasting, as defined in COMET Wang et al. (2023).

**Representation Learning.** For an input EEG sample $\boldsymbol{x}_i$, where $i$ denotes the sample index, we apply two data augmentations $a$ and $b$ to generate two distinct views $\boldsymbol{x}_i^a$ and $\boldsymbol{x}_i^b$. Given a backbone encoder $f(\cdot)$ and a projection head $g(\cdot)$, we first compute intermediate representations via the encoder $\boldsymbol{h}_i^a = f(\boldsymbol{x}_i^a), \boldsymbol{h}_i^b = f(\boldsymbol{x}_i^b)$. These are then passed through the projection head to obtain denser representations $\boldsymbol{z}_i^a = g(\boldsymbol{h}_i^a), \boldsymbol{z}_i^b = g(\boldsymbol{h}_i^b)$ for self-supervised contrastive pre-training.

**Sample-Level Contrasting.** Sample-level contrastive learning treats different augmented views of the same sample as positive pairs, and views from different samples as negative pairs. This instance discrimination framework is widely used in contrastive learning, as demonstrated in SimCLR Chen et al. (2020) and MoCo He et al. (2020), and helps the model learn general representations from unlabeled EEG data Wu et al. (2018). The sample-level contrastive loss $\mathcal{L}_{co}^{sam}$ is defined as:

$$\mathcal{L}_{co}^{sam} = \mathbb{E}_{\boldsymbol{x}_i}\left[-\log\frac{\exp(\mathrm{sim}(\boldsymbol{z}_i^a, \boldsymbol{z}_i^b)/\tau)}{\sum_j \left(\exp(\mathrm{sim}(\boldsymbol{z}_i^a, \boldsymbol{z}_j^b)/\tau)\right)}\right] \tag{3}$$

where $j$ indexes all other samples in the batch $\mathcal{B}$, and $\mathrm{sim}(\boldsymbol{u}, \boldsymbol{v}) = \frac{\boldsymbol{u}^\top \boldsymbol{v}}{\|\boldsymbol{u}\|\|\boldsymbol{v}\|}$ denotes cosine similarity. The temperature parameter $\tau$ controls the sharpness of the distribution.

**Subject-Level Contrasting.** In EEG-based AD detection, each subject is generally associated with a stable pathological brain state in a short period, as discussed in section B. Once a subject is diagnosed with AD or exhibits early signs, all EEG segments from that subject are expected to share similar disease-relevant features. This assumption aligns naturally with subject-level contrasting, a framework defined by Wang et al. (2023) and demonstrated to be effective in various EEG- and ECG-based disease detection tasks Kiyasseh et al. (2021); Abbaspourazad et al. (2024). In this setting, samples from the same subject are considered positive pairs, and those from different subjects are considered negative. The subject-level contrastive loss $\mathcal{L}_{co}^{sub}$ is defined as:

$$\mathcal{L}_{co}^{sub} = \mathbb{E}_{\boldsymbol{x}_i}\left[\mathbb{E}_{\boldsymbol{x}_k}\left[-\log\frac{\exp(\mathrm{sim}(\boldsymbol{z}_i^a, \boldsymbol{z}_k^b)/\tau)}{\sum_j \left(\exp(\mathrm{sim}(\boldsymbol{z}_i^a, \boldsymbol{z}_j^b)/\tau)\right)}\right]\right] \tag{4}$$

where $\boldsymbol{x}_k$ denotes a sample in the batch sharing the same subject ID as $\boldsymbol{x}_i$, i.e., $\boldsymbol{s}_k = \boldsymbol{s}_i$.

**Overall Loss Function.** The total self-supervised pre-training objective combines both contrastive losses as a weighted sum $\mathcal{L}_{Pre} = \lambda_1 \mathcal{L}_{co}^{sam} + \lambda_2 \mathcal{L}_{co}^{sub}$, where $\lambda_1 + \lambda_2 = 1$, and $\lambda_1, \lambda_2 \in [0, 1]$ control the relative importance of each loss component.

## APPENDIX E   DATA AUGMENTATION BANKS

For contrastive pre-training, we employ a bank of data augmentation methods to enhance the model's robustness and generalization capabilities. During the forward pass in the training of each iteration, one augmentation method will be picked from available augmentation options with equal probability. The data augmentation methods include temporal flipping, temporal masking, frequency masking, channel masking, jittering, and dropout, and can be further expanded to more choices.

**1) Temporal Flippling.** We reverse the EEG data along the temporal dimension. The probability of applying this augmentation is controlled by a parameter *prob*, with a default value of 0.5. **2) Temporal Masking.** We randomly mask timestamps across all channels. The proportion of timestamps masked is controlled by the parameter *ratio*, with a default value of 0.1. **3) Frequency Masking.** This method involves converting the EEG data into the frequency domain, randomly masking some frequency bands, and then converting it back. The proportion of frequency bands masked is controlled by the parameter *ratio*, with a default value of 0.1. **4) Channel Masking.** We randomly mask channels across all timestamps. The proportion of channel masked is controlled by the parameter *ratio*, with a default value of 0.1. **5) Jittering.** Random noise, ranging from 0 to 1, is added to the raw data. The intensity of the noise is adjusted by the parameter *scale*, which is set by default to 0.1. **6) Dropout.** Similar to the dropout layer in neural networks, this method randomly drops some values. The proportion of values dropped is controlled by the parameter *ratio*, with a default value of 0.1.

**Table 7: AD/Dementia vs HC Results.** The binary classification task is defined as **HC** vs. **DEM** in CAUEEG and **AD** vs. **HC** in the other three datasets. In CAUEEG, the Dementia category includes mainly AD subjects, with some Vascular Dementia and very mild Parkinson's Disease Dementia. We adhere to this default class split.

| Datasets | ADFTD (257,303 Samples) (65 Subjects, 2 Classes) | | CNBPM (161,845 Samples) (126 Subjects, 2 Classes) | | Cognision (112,860 Samples) (180 Subjects, 2 Classes) | | CAUEEG (1,879,145 Samples) (770 Subjects, 2 Classes) | |
|---|---|---|---|---|---|---|---|---|
| Metrics / Methods | Sensitivity | F1 Score | Sensitivity | F1 Score | Sensitivity | F1 Score | Sensitivity | F1 Score |
| **Sample-Level Classification** | | | | | | | | |
| ManualFeature | 63.51±3.33 | 63.48±3.34 | 65.68±5.00 | 63.78±5.58 | 53.54±1.09 | 53.05±0.99 | 69.90±1.44 | 69.91±1.40 |
| EEGNet | 64.44±4.21 | 64.38±4.16 | 70.83±5.57 | 67.83±7.09 | 52.70±2.39 | 50.70±3.17 | 68.56±1.25 | 68.45±1.15 |
| TST | 71.17±1.53 | 71.01±1.68 | 84.80±5.07 | 83.24±6.24 | 56.34±2.37 | 55.80±2.75 | 73.10±1.91 | 73.24±1.89 |
| EEGInception | **81.40±2.06** | **81.06±2.17** | 86.31±2.17 | 84.32±5.10 | 60.09±1.46 | 59.98±1.34 | 74.78±2.04 | 74.75±1.94 |
| EEGConformer | 79.61±2.72 | 79.49±2.75 | 87.95±3.18 | 85.38±6.17 | 58.69±3.05 | 58.39±3.04 | 75.93±1.67 | 76.09±1.61 |
| BIOT | 73.83±2.81 | 73.70±2.58 | 73.91±4.48 | 73.65±5.20 | 54.37±2.94 | 54.15±2.95 | 73.91±1.89 | 73.99±1.85 |
| MedGNN | 79.41±2.56 | 79.06±2.50 | 85.35±5.22 | 84.57±5.73 | 60.02±2.48 | 59.96±2.50 | 74.79±2.01 | 74.97±1.97 |
| Medformer | 75.10±2.92 | 75.03±3.10 | 85.68±5.04 | 83.63±6.98 | 57.16±2.38 | 56.88±2.21 | 73.44±1.85 | 73.59±1.86 |
| LaBraM | 79.92±5.18 | 79.72±5.27 | 87.30±5.49 | 85.79±6.56 | 61.70±2.95 | 61.40±3.06 | **76.80±1.63** | **77.01±1.59** |
| CBraMod | 71.21±3.65 | 70.96±3.52 | 79.09±4.56 | 78.79±4.84 | 55.26±3.02 | 54.81±3.52 | 72.37±2.19 | 72.50±2.20 |
| **LEAD** | 80.75±4.00 | 80.46±3.96 | **90.93±3.49** | **90.42±3.55** | **62.02±2.15** | **61.87±2.24** | 75.44±1.64 | 75.57±1.60 |
| **Subject-Level Detection** | | | | | | | | |
| ManualFeature | 80.00±5.98 | 80.33±6.23 | 73.08±11.67 | 69.97±15.96 | 53.51±4.32 | 50.44±4.18 | 79.08±1.82 | 79.39±1.84 |
| EEGNet | 78.33±7.75 | 78.04±7.92 | 79.23±6.71 | 78.34±7.84 | 54.59±3.15 | 49.49±6.18 | 75.85±2.50 | 75.87±2.14 |
| TST | 86.25±8.40 | 85.52±8.03 | 84.62±10.03 | 84.39±10.09 | 62.16±7.64 | 60.89±7.98 | 81.38±2.56 | 82.14±2.30 |
| EEGInception | 92.08±2.43 | 91.34±2.81 | 86.92±5.76 | 86.63±6.21 | 63.15±5.33 | 62.95±5.37 | 82.35±2.61 | 82.72±2.28 |
| EEGConformer | 92.08±5.17 | 91.35±5.41 | 87.69±10.15 | 87.64±10.21 | 64.44±5.32 | 63.45±5.34 | **83.51±2.64** | **84.22±2.37** |
| BIOT | 89.17±5.00 | 88.41±5.72 | 83.08±3.92 | 82.80±4.18 | 56.50±4.99 | 53.89±6.81 | 81.57±2.43 | 82.12±2.31 |
| MedGNN | 90.83±3.63 | 89.92±3.56 | 86.92±6.25 | 86.89±6.24 | 64.09±5.04 | 63.84±5.16 | 81.87±3.15 | 82.69±3.04 |
| Medformer | 90.42±3.39 | 89.86±3.51 | 85.38±8.21 | 85.08±8.40 | 55.74±3.14 | 54.20±4.28 | 80.71±2.40 | 81.60±2.31 |
| LaBraM | 90.83±7.05 | 89.96±7.31 | 89.23±4.49 | 89.13±4.58 | 67.29±4.38 | 66.09±4.76 | 82.66±1.71 | 83.50±1.57 |
| CBraMod | 89.58±8.33 | 89.30±8.27 | 86.15±1.88 | 86.04±1.97 | 56.76±6.49 | 55.77±7.53 | 82.34±2.92 | 83.14±2.77 |
| **LEAD** | **93.33±4.64** | **92.78±4.61** | 89.23±5.10 | 89.18±5.11 | **70.62±4.52** | **70.30±4.45** | 82.71±1.76 | 83.53±1.72 |

# APPENDIX F  SUPPLEMENTARY EXPERIMENTS

## F.1  AD/DEMENTIA VS HC CLASSIFICATION

*AD/Dementia vs. HC Classification.* We next perform a binary classification task to evaluate the model's ability to distinguish AD subjects from HC. AD is the most prevalent neurodegenerative disorder among older adults, and differentiating AD from HC enables the potential identification of AD-related features compared with normal aging. Such distinctions are critical for early-stage EEG-based AD detection and may further support integration with wearable devices for at-home monitoring, as discussed in the introduction. For the CAUEEG dataset, we follow the default class split, in which AD is grouped under the broader Dementia (DEM) category. In contrast, the other three datasets provide explicit AD versus HC labels.

The results are presented in Table 7. Our method achieves the highest scores in 13 out of 16 metrics. Among the 4 datasets, ADFTD yields the best subject-level performance. Although ADFTD has the smallest number of subjects, its average per-subject recording length (0.303 h) is the longest, while Cognision has the shortest (0.050 h). This suggests that achieving strong subject-level performance in EEG-based AD detection may require sufficiently long per-subject recordings. Furthermore, since the Cognision dataset was collected initially with only 7 channels, it theoretically contains less spatial information compared to the other datasets. The combination of shorter recording length per subject and fewer channels likely contributes to Cognision's relatively limited performance.

## F.2  LEAVE-ONE-SUBJECT-OUT ANALYSIS

We evaluate our method under the leave-one-subject-out (LOSO) setting on the widely used benchmark dataset ADFTD. This setup enables us to assess detection performance at the individual subject level and examine how demographic factors, Mini-Mental State Examination (MMSE) scores, and related variables affect model performance. Specifically, each subject is iteratively held out for testing while all remaining subjects are used for training. Since no validation set is available here, we disable early stopping to avoid information leakage and fix the training to 30 epochs. We choose sample-level accuracy and subject-level accuracy as metrics, as the test set consists of only one subject (one class).

**Table 8: First Part of LOSO results of ADFTD.** Demographic information, MMSE, recording length, and results of all AD and HC subjects from Sub-001 to Sub-065.

| Subject ID | Gender | Age | Class | MMSE | Length | Sample-level Accuracy | Subject-level Accuracy |
|---|---|---|---|---|---|---|---|
| Sub-001 | F | 57 | AD | 16 | 0.25h | 76.37% | ✓ |
| Sub-002 | F | 78 | AD | 22 | 0.33h | 53.30% | ✓ |
| Sub-003 | M | 70 | AD | 14 | 0.16h | 88.35% | ✓ |
| Sub-004 | F | 67 | AD | 20 | 0.29h | 98.30% | ✓ |
| Sub-005 | M | 70 | AD | 22 | 0.30h | 98.30% | ✓ |
| Sub-006 | F | 61 | AD | 14 | 0.28h | 90.56% | ✓ |
| Sub-007 | F | 79 | AD | 20 | 0.33h | 56.30% | ✓ |
| Sub-008 | M | 62 | AD | 16 | 0.33h | 79.61% | ✓ |
| Sub-009 | F | 77 | AD | 23 | 0.28h | 55.18% | ✓ |
| Sub-010 | M | 69 | AD | 20 | 0.44h | 29.08% | ✗ |
| Sub-011 | M | 71 | AD | 22 | 0.29h | 88.62% | ✓ |
| Sub-012 | M | 63 | AD | 18 | 0.34h | 78.17% | ✓ |
| Sub-013 | F | 64 | AD | 20 | 0.36h | 88.84% | ✓ |
| Sub-014 | M | 77 | AD | 14 | 0.40h | 68.71% | ✓ |
| Sub-015 | M | 61 | AD | 18 | 0.33h | 86.84% | ✓ |
| Sub-016 | F | 68 | AD | 14 | 0.39h | 89.41% | ✓ |
| Sub-017 | F | 61 | AD | 6 | 0.35h | 98.22% | ✓ |
| Sub-018 | F | 73 | AD | 23 | 0.34h | 98.32% | ✓ |
| Sub-019 | F | 62 | AD | 14 | 0.34h | 80.42% | ✓ |
| Sub-020 | M | 71 | AD | 4 | 0.35h | 93.02% | ✓ |
| Sub-021 | M | 79 | AD | 22 | 0.30h | 65.15% | ✓ |
| Sub-022 | F | 68 | AD | 20 | 0.30h | 96.93% | ✓ |
| Sub-023 | M | 60 | AD | 16 | 0.25h | 35.84% | ✗ |
| Sub-024 | F | 69 | AD | 20 | 0.23h | 98.91% | ✓ |
| Sub-025 | F | 79 | AD | 20 | 0.26h | 32.68% | ✗ |
| Sub-026 | F | 61 | AD | 18 | 0.27h | 98.21% | ✓ |
| Sub-027 | F | 67 | AD | 16 | 0.26h | 99.64% | ✓ |
| Sub-028 | M | 49 | AD | 20 | 0.33h | 88.33% | ✓ |
| Sub-029 | F | 53 | AD | 16 | 0.23h | 99.41% | ✓ |
| Sub-030 | F | 56 | AD | 20 | 0.18h | 95.70% | ✓ |
| Sub-031 | F | 67 | AD | 22 | 0.34h | 56.23% | ✓ |
| Sub-032 | F | 59 | AD | 20 | 0.24h | 93.16% | ✓ |
| Sub-033 | F | 72 | AD | 20 | 0.22h | 91.41% | ✓ |
| Sub-034 | F | 75 | AD | 18 | 0.29h | 88.98% | ✓ |
| Sub-035 | F | 57 | AD | 22 | 0.32h | 91.02% | ✓ |
| Sub-036 | F | 58 | AD | 9 | 0.26h | 78.26% | ✓ |
| Sub-037 | M | 57 | HC | 30 | 0.37h | 88.61% | ✓ |
| Sub-038 | M | 62 | HC | 30 | 0.35h | 89.40% | ✓ |
| Sub-039 | M | 70 | HC | 30 | 0.34h | 95.36% | ✓ |
| Sub-040 | M | 61 | HC | 30 | 0.32h | 96.62% | ✓ |
| Sub-041 | F | 77 | HC | 30 | 0.34h | 88.48% | ✓ |
| Sub-042 | M | 74 | HC | 30 | 0.33h | 68.55% | ✓ |
| Sub-043 | M | 72 | HC | 30 | 0.31h | 98.10% | ✓ |
| Sub-044 | F | 64 | HC | 30 | 0.38h | 92.06% | ✓ |
| Sub-045 | F | 70 | HC | 30 | 0.34h | 99.10% | ✓ |
| Sub-046 | M | 63 | HC | 30 | 0.36h | 90.99% | ✓ |
| Sub-047 | F | 70 | HC | 30 | 0.34h | 79.91% | ✓ |
| Sub-048 | M | 65 | HC | 30 | 0.38h | 95.42% | ✓ |
| Sub-049 | F | 62 | HC | 30 | 0.34h | 97.24% | ✓ |
| Sub-050 | M | 68 | HC | 30 | 0.31h | 97.35% | ✓ |
| Sub-051 | F | 75 | HC | 30 | 0.33h | 99.71% | ✓ |
| Sub-052 | F | 73 | HC | 30 | 0.33h | 91.46% | ✓ |
| Sub-053 | M | 70 | HC | 30 | 0.31h | 95.40% | ✓ |
| Sub-054 | M | 78 | HC | 30 | 0.35h | 89.72% | ✓ |
| Sub-055 | M | 67 | HC | 30 | 0.33h | 96.22% | ✓ |
| Sub-056 | F | 64 | HC | 30 | 0.34h | 97.27% | ✓ |
| Sub-057 | M | 64 | HC | 30 | 0.32h | 97.77% | ✓ |
| Sub-058 | M | 62 | HC | 30 | 0.27h | 95.02% | ✓ |
| Sub-059 | M | 77 | HC | 30 | 0.30h | 46.82% | ✗ |
| Sub-060 | F | 71 | HC | 30 | 0.33h | 99.07% | ✓ |
| Sub-061 | F | 63 | HC | 30 | 0.33h | 50.85% | ✓ |
| Sub-062 | M | 67 | HC | 30 | 0.35h | 78.71% | ✓ |
| Sub-063 | M | 66 | HC | 30 | 0.34h | 99.29% | ✓ |
| Sub-064 | M | 66 | HC | 30 | 0.35h | 99.90% | ✓ |
| Sub-065 | F | 71 | HC | 30 | 0.30h | 99.72% | ✓ |

**Table 9: Second Part of LOSO results of ADFTD.** Demographic information, MMSE, recording length, and results of all FTD subjects from Sub-066 to Sub-088. The average in the last line applies to all subjects in the two tables, from Sub-001 to Sub-088.

| Subject ID | Gender | Age | Class | MMSE | Length | Sample-level Accuracy | Subject-level Accuracy |
|---|---|---|---|---|---|---|---|
| Sub-066 | M | 73 | FTD | 20 | 0.22h | 81.99% | ✓ |
| Sub-067 | M | 66 | FTD | 24 | 0.28h | 98.08% | ✓ |
| Sub-068 | M | 78 | FTD | 25 | 0.22h | 28.16% | ✗ |
| Sub-069 | M | 70 | FTD | 22 | 0.31h | 98.08% | ✓ |
| Sub-070 | F | 67 | FTD | 22 | 0.25h | 46.22% | ✗ |
| Sub-071 | M | 62 | FTD | 20 | 0.31h | 37.67% | ✓ |
| Sub-072 | M | 65 | FTD | 18 | 0.28h | 55.27% | ✓ |
| Sub-073 | F | 57 | FTD | 22 | 0.39h | 89.56% | ✓ |
| Sub-074 | F | 53 | FTD | 20 | 0.37h | 89.45% | ✓ |
| Sub-075 | F | 71 | FTD | 22 | 0.31h | 75.56% | ✓ |
| Sub-076 | M | 44 | FTD | 24 | 0.35h | 81.74% | ✓ |
| Sub-077 | M | 61 | FTD | 22 | 0.27h | 71.81% | ✓ |
| Sub-078 | M | 62 | FTD | 22 | 0.34h | 90.40% | ✓ |
| Sub-079 | F | 60 | FTD | 18 | 0.25h | 95.25% | ✓ |
| Sub-080 | F | 71 | FTD | 20 | 0.28h | 92.25% | ✓ |
| Sub-081 | F | 61 | FTD | 18 | 0.25h | 13.84% | ✗ |
| Sub-082 | M | 63 | FTD | 27 | 0.24h | 41.68% | ✓ |
| Sub-083 | F | 68 | FTD | 20 | 0.27h | 79.62% | ✓ |
| Sub-084 | F | 71 | FTD | 24 | 0.20h | 73.19% | ✓ |
| Sub-085 | M | 64 | FTD | 26 | 0.18h | 62.65% | ✓ |
| Sub-086 | M | 49 | FTD | 26 | 0.18h | 23.24% | ✗ |
| Sub-087 | M | 73 | FTD | 24 | 0.19h | 78.71% | ✓ |
| Sub-088 | M | 55 | FTD | 24 | 0.25h | 89.32% | ✓ |
| Average | – | 66.17 | – | 22.94 | 0.30h | 80.84% | 90.91% |

All other experimental settings and parameters remain unchanged. The detailed results are presented in Table 8 and Table 9.

# APPENDIX G    MORE ABLATION STUDY

## G.1    PRE-TRAINING DATASETS ABLATION STUDY

We conduct an ablation study to evaluate the effectiveness of each domain-relevant pre-training dataset. Starting from fully supervised learning on ADFTD, we progressively enable pre-training by adding two datasets at a time. The results are shown in Table 10. Both sample-level and subject-level F1 scores improve steadily as more datasets are included. Nearly all pre-training datasets contribute positively, either to sample-level or subject-level performance. The only exceptions are BrainLat and Depression, which lead to performance drops.

We suspect this may be due to data quality issues, such as excessive artifacts in certain subjects, which can introduce noise and degrade model performance. This points to a promising direction for future work: developing automated pipelines for filtering low-quality datasets and subjects. Overall, our domain-relevant dataset selection strategy, which focuses on resting-state recordings from both healthy

**Table 10: Ablation Study of Pre-training Datasets.** Our full model is trained on 12 pre-training datasets and fine-tuned on ADFTD. We begin with fully supervised learning on ADFTD alone, and then progressively enable pre-training by adding two datasets at a time.

| Models | Sample-Level F1 Score | Subject-Level F1 Score |
|---|---|---|
| Fully Supervised Learning on ADFTD | 59.08±5.19 | 70.20±8.73 |
| + Pre-training on AD-Auditory, BACA-RS | 61.84±3.61 ↑2.76 | 69.98±7.70 ↓0.22 |
| + BrainLat, Depression | 61.82±3.46 ↓0.02 | 66.16±10.60 ↓3.82 |
| + FEPCR, MCEF-RS | 62.58±2.89 ↑0.76 | 69.24±7.14 ↑3.08 |
| + P-ADIC, PD-RS | 64.34±4.34 ↑1.76 | 70.00±7.19 ↑0.76 |
| + PEARL-Neuro, SRM-RS | 63.86±3.29 ↓0.48 | 75.09±8.16 ↑5.09 |
| + TDBrain, TUEP = LEAD | 67.41±3.77 ↑3.5 | 75.77±5.81 ↑0.68 |

subjects and patients with neurological disorders, proves effective. Almost all datasets contribute to performance gains, and the final model achieves the highest overall accuracy with only 730.48 hours of pre-training data, surpassing other EEG foundation models trained on much larger corpora.

## APPENDIX H   DISCUSSION

### H.1   LIMITATIONS AND FUTURE WORKS

In this paper, we demonstrate the effectiveness of large-scale training and propose a subject-level AD detection pipeline. Nevertheless, several limitations remain. First, while we achieve strong performance on ADFTD under the LOSO setting, reaching 90.91% accuracy in distinguishing AD, FTD, and HC subjects, and similarly high accuracy (around 90%) in distinguishing AD/DEM from HC, the performance for separating AD, MCI, and HC is still limited at around 60%. This highlights the continued challenge of early-stage detection. A likely explanation lies in the overlapping distribution of dementia categories. Dementia is a progressive neurodegenerative condition: healthy older adults may transition into MCI, and some MCI cases eventually progress to AD. However, not all MCI cases advance; some remain stable (SMCI) or even revert to normal cognition, while others progress (PMCI) Vecchio et al. (2018); Ge et al. (2025). Critically, many PMCI patients may already exhibit AD-related features at the time of data acquisition, although these are not identifiable without longitudinal follow-up, while some SMCI patients are closer to HC. This ambiguity makes certain MCI subjects difficult to distinguish from either AD or HC. Future work should therefore prioritize collecting longitudinal data to track disease progression and subject-level shifts.

Second, from a methodological perspective, the pipeline can be further enhanced by incorporating more domain-relevant pre-training datasets, designing stronger backbone encoders, and adopting more advanced self-supervised pre-training strategies. For example, our current channel alignment strategy provides the advantage of avoiding sequential pre-training on datasets with different channel topologies, which could otherwise lead to catastrophic forgetting. Nevertheless, this step could be replaced by employing a cross-attention mechanism to automatically map additional channels onto the 19 standard channels, thereby replacing manual alignment. In addition, masked reconstruction objectives could be combined with the existing contrastive pre-training framework to capture more effective multi-scale and context-aware representations. Theoretically, with richer datasets and more sophisticated training methodologies, detection performance could be further improved.

## APPENDIX I   IMPLEMENTATION DETAILS

**Manual Feature** utilize 32 features, including mean, variance, skewness, kurtosis, std, iqr, max, min, mean, median, delta power, theta power, alpha power, beta power, total power, theta alpha ratio, alpha beta ratio, delta relative power, theta relative power, alpha relative power, beta relative power, phase coherence, spectral centroid, spectral rolloff, spectral peak, average magnitude, median frequency, amplitude modulation, spectral entropy, tsallis entropy, and shannon entropy. A linear layer is then applied to perform final classification.

**EEGNet** is a classic deep learning method for EEGNet decoding. It uses depthwise and separable convolutions to capture spatial and temporal features. We keep the same structure and order when applying convolutions, normalization, and activations as described in the paper.

**TST** is introduced to apply the transformer to time series. It embeds each cross-channel timestamp as an input token for self-attention. We set $e\_layers$ = 12, $n\_heads$ = 8, $d\_model$ = 128, and $d\_ff$ = 256.

**EEGInception** uses different scales of convolutional kernels, combined with a spatial block for feature extraction. We set $n\_blocks$ = 3, $channels$ = (96,192,384), $kernel\_sizes$ = (8,16,32), $depth\_multiplier$ = 2, $bottleneck\_channels$ = 32.

**EEGConformer** uses convolutional modules to learn low-level local features and embed the raw data into patches for self-attention. We set $e\_layers$ = 12, $n\_heads$ = 8, $d\_model$ = 128, and $d\_ff$ = 256.

**BIOT** employs single-channel patch embedding to handle biosignals with varying numbers of channels. Each patch is mapped into tokens, with segment embedding, channel embedding, and positional embedding added to make the tokens distinguishable. We set $e\_layers$ = 12, $n\_heads$ = 8, $d\_model$ = 128, and $d\_ff$ = 256.

**MedGNN** is a graph-nerual-network-based method for medical time-series classification. It uses a multi-resolution graph transformer architecture to model the dynamic dependencies and fuse the

information from different resolutions. We set *–resolution_list* = 2,4,6,8, *–nodedim* = 10, *e_layers* = 12, *n_heads* = 8, *d_model* = 128, and *d_ff* = 256.

**Medformer** is designed for biomedical time series classification, including EEG and ECG. Cross-channel multi-granularity patch embedding and intra-inter-granularity self-attention are utilized. We extended our work based on this method. We set *e_layers* = 12, *d_model* = 128, and *d_ff* = 256, *patch_len_list* = [2, 4, 8].

**LaBraM.** LaBraM is the first EEG foundation model, trained on 2,000 hours of EEG recordings collected from multiple datasets. Its pre-training follows a two-step strategy that combines vector quantization with mask-based reconstruction. In our experiments, we use their released pre-trained checkpoint with default parameters and fine-tune it on four AD downstream datasets. To match the channel configuration of our datasets to their checkpoint, we apply a 1D convolution (Conv1D) for channel mapping. We retain the default architectural parameters from the checkpoint and set *batch_size* = 128 and *segment_length* = 400.

**CBraMod.** CBraMod Wang et al. (2025) is a state-of-the-art EEG foundation model trained on the 9000 hours TUEG dataset with 19 standard channels of the 10-20 system. They use a criss-cross transformer to perform spatial and temporal attention to capture features along two dimensions. We keep their default parameters to load their pre-trained checkpoint and fine-tune on 4 AD downstream datasets. Specifically, their parameters are set to *in_dim* = 200, *out_dim* = 200, *d_model* = 200, *dim_feedforward* = 800, *seq_len* = 30, *n_layer* = 12, *nhead* = 8.

**LEAD (Ours).** We pre-train on 12 (3 Ad and 9 Non-AD) datasets and then fine-tune on the 4 downstream AD datasets. We set the patch length $L$ to 4, and the target channel number $F$ to 76. The sample- and subject-level cross-entropy loss weights $\alpha$ and $\beta$ are both set to 0.5. The contrastive loss coefficients $\lambda_1$ and $\lambda_2$ are set to 0.25 and 0.75. We set *e_layers* = 12, *n_heads* = 8, *d_model* = 128, *d_ff* = 256, and *group_size* = 4.

## APPENDIX J    DATASETS PREPROCESSING

Our preprocessing pipeline involves the following steps, applied sequentially:

**1) Artifact Removal:** For datasets lacking prior artifact rejection, we utilize independent component analysis (ICA) combined with the ICLabel algorithm Pion-Tonachini et al. (2019) to automatically identify and remove components associated with artifacts like eye blinks, or muscle activity.

**2) Channel Alignment:** We align all datasets to the standard 19-channel montage based on the international 10-20 system: Fp1, Fp2, F7, F3, Fz, F4, F8, T3/T7, C3, Cz, C4, T4/T8, T5/P7, P3, Pz, P4, T6/P8, O1, and O2 Homan et al. (1987). If a dataset has fewer channels, missing ones are interpolated using spherical splines (via the MNE package Gramfort et al. (2013)) based on 3D electrode coordinates. If a dataset has more channels, only these 19 are selected based on their names, and the rest are discarded. For datasets employing different montages (e.g., Biosemi bio), signals are projected onto the target 19 channels using their 3D coordinates. As most EEG datasets include at least these 19 standard channels, alignment enables the model to train and test seamlessly across datasets with heterogeneous topologies.

**5) Band-Pass Filtering:** Each trial undergoes band-pass filtering between 0.5 Hz and 45 Hz. This step removes slow drifts and high-frequency noise, which are generally outside the frequency range of scalp-recorded brain activity.

**3) Frequency Alignment:** All datasets are resampled to a uniform sampling frequency of 128 Hz. This rate is widely used, adequately captures the main physiological EEG frequency bands ($\delta$, $\theta$, $\alpha$, $\beta$, $\gamma$), and reduces high-frequency noise.

**4) Data Segmentation:** We newly propose a **multi-scale segmentation** strategy. In addition to the 128 Hz signals, trials are further downsampled to 64 Hz and 32 Hz. The 128 Hz, 64 Hz, and 32 Hz trials are then segmented into half-overlapping 128-timestep windows, corresponding to 1s, 2s, and 4s samples, respectively. This inherently enables multi-scale feature extraction and increases the number of samples per subject, thereby benefiting majority voting. A sampling rate label $r$ is assigned to each sample for subsequent use.

**6) Z-Score Normalization:** Finally, Z-score normalization is applied to each segmented sample, computed independently for each channel. Since our study does not involve ERP datasets and most training resources are resting-state or paradigms closely aligned with resting state, normalization does not disrupt feature-related information.

The following subsections provide detailed data preprocessing steps for each dataset.

## J.1  AD DATASETS

### J.1.1  AD-AUDITORY.

The AD-Auditory (40Hz Auditory Entrainment) is a publicly available EEG dataset on the Open-NEURO website[1] from the paper Lahijanian et al. (2024). It contains 35 subjects, including 17 AD, 6 MCI, 10 healthy controls, and 2 unknown subjects. This dataset aims to investigate the effect of entrainment on brain oscillations using EEG signal recordings during auditory brain stimulation for distinguish Alzheimer's Disease. All the data are recorded using 19 monopolar channels (Fp1, Fp2, F7, F3, Fz, F4, F8, T3, C3, Cz, C4, T4, T5, P3, Pz, P4, T6, O1, and O2) based on the standard 10/20 system, with a sampling rate set to 250Hz. The dataset's authors preprocess the data using the EEGLab toolbox in Matlab, which includes bandpass filtering, noise removal, artifact removal, re-referencing, and interpolating rejected channels, as described in their paper and on the data website. For consistency with our pipeline, we performed additional preprocessing following the unified steps.

### J.1.2  ADFTD.

The ADFTD-RS (A dataset of EEG recordings from Alzheimer's disease, Frontotemporal dementia and Healthy subjects) is a publicly available resting-state EEG dataset on the OpenNEURO website[2] from the paper Miltiadous et al. (2023b;a), and a complementary dataset ADFTD-PS(A complementary dataset of open-eyes EEG recordings in a photo-stimulation setting from: Alzheimer's disease, Frontotemporal dementia and Healthy subjects) in a photo-stimulation setting with exactly matched subjects[3]. It contains 88 subjects, including 36 AD, 23 Frontotemporal Dementia (FTD), and 29 healthy controls. For recording, a Nihon Kohden EEG 2100 clinical device is used, with 19 scalp electrodes (Fp1, Fp2, F7, F3, Fz, F4, F8, T3, C3, Cz, C4, T4, T5, P3, Pz, P4, T6, O1, and O2) according to the 10-20 international system and 2 reference electrodes (A1 and A2) placed on the mastoids for impedance check, according to the manual of the device. Each recording is performed according to the clinical protocol, with participants sitting with their eyes closed or in photo-stimulation setting. The collection sampling rate is 500Hz. The dataset's authors preprocess the data using the EEGLab toolbox in Matlab, which includes bandpass filtering, noise removal, artifact removal, re-referencing, and interpolating rejected channels, as described in their paper and on the data website. Since their preprocessing pipeline is already comprehensive, we only perform additional steps to align with our unified pipeline: downsampling to 128 Hz, applying multi-scale segmentation, and Z-score normalization. Finally, we concatenated ADFTD-RS and ADFTD-PS into a single dataset (ADFTD) based on subject IDs.

### J.1.3  BRAINLAT.

The BrainLat[4] (Latin American Brain Health Institute) dataset comprises multimodal neuroimaging data from 780 participants from Latin America Prado et al. (2023). It contains two modalities: EEG and MRI. It includes five classes of subjects: Alzheimer's disease (AD), behavioral variant frontotemporal dementia (bvFTD), multiple sclerosis (MS), Parkinson's disease (PD), and healthy controls (HC). For EEG data recording, subjects are recorded in an eye-closed resting state inside a dimly lit, sound-attenuated, and electromagnetically shielded EEG room. They are instructed to remain still and awake, with a 128-channel Biosemi Active-two acquisition system (pin-type, active, sintered Ag-AgCl electrodes). The data are band-pass filtered between 0.5 and 40 Hz using a zero-phase shift Butterworth filter of order 8. The data are then downsampled to 512 Hz, and

---

[1]https://openneuro.org/datasets/ds005048/versions/1.0.0

[2]https://openneuro.org/datasets/ds004504/versions/1.0.8

[3]https://openneuro.org/datasets/ds006036/versions/1.0.5

[4]https://www.synapse.org/Synapse:syn51549340/wiki/624187

Independent Component Analysis (ICA) is used to correct EEG artifacts induced by blinking and eye movements. We perform secondary preprocessing to match the pipeline of our method.

EEG data for each subject are stored in folders labeled AR and CL, representing the subjects' countries: Argentina and Chile. It is important to note that some subjects cannot read for unknown reasons, such as the subject named "sub-100013" (at least when we downloaded the dataset, which the data version was last modified by Dr. Pavel Prado on 7/2/2024). Additionally, not all subjects have EEG data; most subjects only have MRI datasets. In total, there are 135 functional subjects with EEG data across all five classes. Recordings were acquired using a BioSemi 128-channel system, which differs substantially from the standard 10–20 montage in electrode naming and placement. To ensure consistency across datasets, we performed channel alignment with the MNE-Python toolbox Gramfort et al. (2013), mapping the 128 BioSemi electrodes to the 19 standard 10–20 channels using their 3D coordinates. All subsequent preprocessing steps followed the same pipeline.

### J.1.4 CNBPM.

The CNBPM is a large private EEG dataset provided by the AI-LAB laboratory at the University Mediterranea of Reggio Calabria, Italy, referenced in studies Ieracitano et al. (2019b); Amezquita-Sanchez et al. (2019). It consists of 63 subjects with Alzheimer's Disease (AD), 63 with Mild Cognitive Impairment (MCI), and 63 Healthy Control (HC) subjects. The data are collected using 19 standard channels (Fp1, Fp2, F7, F3, Fz, F4, F8, T3, C3, Cz, C4, T4, T5, P3, Pz, P4, T6, O1, and O2) with an initial sampling rate of 1024Hz. A frequency-band filter is applied to filter the frequency bands between 0.5 and 32 Hz, followed by downsampling to reduce the sampling rate to 256Hz. Visible blinks affected by artifacts are visually inspected and removed by an EEG expert. Since their preprocessing pipeline is already comprehensive, we only perform additional steps to align with our unified pipeline: downsampling to 128 Hz, applying multi-scale segmentation, and Z-score normalization.

### J.1.5 COGNISION.

The Cognision is a private EEG dataset from the *Cognision* company that consists of resting-state EEG data. It contains 180 subjects, including 97 Alzheimer's Disease (AD) subjects and 83 Healthy Control (HC) subjects. Each subject has a recording trial with 22,524 timestamps collected at 125Hz. The number of channels is 7 (Fz, Cz, Pz, F3, P3, F4, P4). Artifacts, such as eye movements, are visually inspected and removed by an EEG expert. To ensure compatibility with our preprocessing pipeline, we performed secondary processing. Specifically, the available 7 channels were used to interpolate the remaining 12 channels (Fp1, Fp2, F7, F8, T3, C3, C4, T4, T5, T6, O1, O2), resulting in the full set of 19 electrodes defined in the international 10-20 system via 3D coordinate mapping. All subsequent preprocessing steps followed the same unified pipeline.

### J.1.6 P-ADIC

The P-ADIC dataset, introduced in Shor et al. (2021) and publicly available via DRYAD[5], includes EEG recordings from 249 subjects (although the original paper reports 230). The dataset includes 49 individuals with Alzheimer's Disease (AD), 34 with Mild Cognitive Impairment (MCI), 96 Healthy Controls (HC), 42 with Schizophrenia, and 28 with Depression. EEG signals are recorded using 19 standard channels (Fp1, Fp2, F7, F3, Fz, F4, F8, T3, C3, Cz, C4, T4, T5, P3, Pz, P4, T6, O1, O2) at an initial sampling rate of 500 Hz. As no preprocessing steps (e.g., artifact removal) are reported in the original publication, we preprocess the raw data from scratch, using the same unified preprocessing pipeline described before.

### J.1.7 CAUEEG

The CAUEEG dataset, introduced in Kim et al. (2023), is available upon request and contains 1,379 EEG recordings from 1,155 subjects. It includes 459 recordings from healthy controls (HC), 417 from individuals with Mild Cognitive Impairment (MCI), 311 with dementia, and 192 with other conditions. EEG signals were recorded using 19 standard channels (Fp1, F3, C3, P3, O1, Fp2, F4, C4, P4, O2, F7, T3, T5, F8, T4, T6, Fz, Cz, Pz) at a sampling rate of 200 Hz and pre-filtered with

---

[5]https://datadryad.org/dataset/doi:10.5061/dryad.8gtht76pw

a 0.5–70 Hz bandpass filter. Data collection includes resting-state, photic stimulation, and limited hyperventilation tasks. We apply additional preprocessing to align with our pipeline.

In line with the protocol described in Kim et al. (2023), we treat each EEG recording as an independent subject. This decision is motivated by reported label shifts across sessions (e.g., HC to MCI, MCI to AD), which suggest that different recordings from the same individual should be considered distinct entities. Since this dataset provides detailed event annotations, we perform artifact removal via segment rejection rather than applying ICLabel. Specifically, segments labeled with terms such as 'artifact', 'blink', 'eye', 'noise', 'movement', 'chewing', 'talk', and related descriptors are removed, together with a 1-second buffer before and after each annotated event. The other steps are the same.

### J.2  NON-AD DATASETS

#### J.2.1  BACA-RS.

The BACA-RS (Resting-state EEG data before and after cognitive activity across the adult lifespan and a 5-year follow-up) dataset is a publicly available EEG dataset on the OpenNEURO website[6], referenced in the paper Getzmann et al. (2024). According to the paper's description, this dataset consists of 64 channels based on the 10-20 system, with the FCz electrode as an online reference. It includes resting-state EEG recordings from 608 subjects aged between 20 and 70 years, along with follow-up measurements of 208 subjects approximately 5 years later, starting in 2021. The EEG data are recorded with eyes open and eyes closed before and after a 2-hour block of cognitive experimental tasks. The EEG data are recorded at a 1000Hz sampling rate and filtered online using a 250Hz low-pass filter. This dataset aims to study the aging of brain activity in a resting state and provide a normal distribution of healthy subjects' resting-state EEG for comparison with clinically relevant disorders. We use the same unified preprocessing pipeline described before.

#### J.2.2  DEPRESSION.

The Depression (EEG: Depression rest) dataset is a publicly available EEG dataset on the Open-NEURO website[7] from the paper Cavanagh et al. (2019); Cavanagh (2021). It contains data from 122 college-age subjects with healthy and different degrees of depression. The EEG data are recorded in a resting state, with instructions for eyes open and eyes closed, triggering one-minute spans of either open or closed eyes. Each subject's depression level is labeled based on their score on the Beck Depression Inventory (BDI). The raw data sampling frequency is 500Hz. We use the same unified preprocessing pipeline described before.

#### J.2.3  FEPCR.

The FEPCR dataset (EEG: First Episode Psychosis vs. Control Resting Task 1 & Task 2) is a publicly available EEG dataset on the OpenNEURO website[8][9] from the paper Phalen et al. (2020). It contains a total of 143 healthy and First-Episode Psychosis (FEP) subjects, with EEGs recorded using a 60-channel, low-impedance 10-10 system cap. We use the same unified preprocessing pipeline.

### J.3  MCEF-RS.

The MCEF-RS dataset (EEG Resting-state Microstates Correlates of Executive Functions) is a publicly available EEG dataset on the OpenNEURO website[10] from the paper Chenot et al. (2024). This study aimed to specifically explore the relationship between intrinsic brain spatio-temporal dynamics and Executive Functions. To do so, resting-state EEG microstates were used to assess brain spatio-temporal dynamics in 140 healthy participants, while a comprehensive battery of nine cognitive function tasks was employed to evaluate their executive functions. We use the same unified preprocessing pipeline.

---

[6]https://openneuro.org/datasets/ds005385/versions/1.0.2
[7]https://openneuro.org/datasets/ds003478/versions/1.1.0
[8]https://openneuro.org/datasets/ds003944/versions/1.0.1
[9]https://openneuro.org/datasets/ds003947/versions/1.0.1
[10]https://openneuro.org/datasets/ds005305/versions/1.0.1

### J.3.1 PD-RS.

The PD-RS (Rest eyes open) dataset is a publicly available EEG dataset on the OpenNEURO website[11], referenced in the paper Singh et al. (2023). This dataset includes 149 subjects, with 100 Parkinson's disease (PD) subjects and 49 Healthy controls (HC) subjects. According toe the description in their paper, the EEG data is recorded with a 64-channel BrainVision cap in a resting state with their eyes open for two minutes. The sampling frequency is set to 500Hz, and a 0.1Hz high-pass filter is applied to the EEG recordings. The Fully Automated Statistical Thresholding for EEG artifact Rejection (FASTER) algorithm rejects the bad channels and trials with greater than +/- 3 Z-scores on key metrics and pop_rejchan function from EEGLAB. Bad channels are interpolated except the mid-frontal Cz channel, which is never interpolated. Eye blink artifacts are removed following ICA. We use the same unified preprocessing pipeline described before.

### J.3.2 PEARL-NEURO.

The PEARL-Neuro (A Polish Electroencephalography, Alzheimer's Risk-genes, Lifestyle and Neuroimaging) dataset is a publicly available EEG dataset on the OpenNEURO website[12], referenced in the paper Dzianok & Kublik (2024). The full dataset contains data from 192 self-reported healthy middle-aged (50-63) subjects, with a balanced female-to-male ratio. Of these, 79 subjects are publicly available, and the dataset includes two modalities: EEG and fMRI. Other information, such as blood tests, demographics, and other health conditions, are also provided. The dataset aims to identify genetic variations associated with brain anatomical and functional phenotype imaging genomics, which could be potential biomarkers for predicting the risk of developing neurological and psychiatric disorders. This could lead to earlier diagnoses, more targeted treatments, and improved patient outcomes. EEG data are recorded using Brain Products systems, including an actiCHamp amplifier and high-density actiCAP electrode caps with 128 electrodes (Brain Products GmbH, Munich, Germany). The FCz electrode is used as an online reference, and the sampling rate is set to 1000Hz with a low-pass filter at 280Hz. The dataset includes three different tasks: the Sternberg memory task (Sternberg), the Multi-source interference task (MSIT), and resting-state (rest). We perform secondary preprocessing to match the pipeline of our method. We only take the resting-state trials from each subject and apply our unified preprocessing pipeline.

### J.3.3 SRM-RS.

The REEG-SRM (SRM Resting-state EEG) dataset is a publicly available EEG dataset on the OpenNEURO website[13], referenced in the paper Hatlestad-Hall et al. (2022). This dataset contains resting-state EEG extracted from the experimental paradigm used in the Stimulus-Selective Response Modulation (SRM) project at the Department of Psychology, University of Oslo, Norway. The EEG data are recorded using 64 electrodes with a BioSemi ActiveTwo system, following the positional scheme of the 10-10 system. The dataset includes 111 healthy control subjects, with some subjects having one trial and others having multiple trials. The sampling rate is set to 1024Hz. Preprocessing steps are applied to the raw data, including bad channel interpolation, artifact rejection, and bandpass filtering from 1Hz to 45Hz. We exclude two subjects who cannot read, identified as "sub-029" and "sub-104." For the remaining 109 subjects, we perform the same unified preprocessing pipeline described before.

### J.3.4 TDBRAIN.

The TDBrain (Two Decades-Brainclinics Research Archive for Insights in Neurophysiology) dataset[14][15], referenced in the paper Van Dijk et al. (2022), is a large permission-available EEG time series dataset recording brain activities of 1274 subjects with 33 channels. Researchers need to send requests to the authors by filling out the application forms to get access to this dataset. This dataset aims to research neurological or psychiatric dysfunction, such as Major Depressive Disorder (MDD), attention deficit hyperactivity disorder (ADHD), Subjective Memory Complaints (SMC),

---

[11]https://openneuro.org/datasets/ds004584/versions/1.0.0
[12]https://openneuro.org/datasets/ds004796/versions/1.0.9
[13]https://openneuro.org/datasets/ds003775/versions/1.2.1
[14]https://brainclinics.com/resources/
[15]https://www.synapse.org/Synapse:syn25671079/wiki/610278

obsessive-compulsive disorder (OCD), Parkinson's disease (PD), and many other brain disorders. The EEG data is recorded in resting-states in eye-open and eye-closed states. The sampling rate is 500Hz. Preprocessing steps are applied to the raw data, including artifact rejection, 50Hz notch-frequency removal, and bandpass filtering from 0.5Hz to 100Hz. we perform the same unified preprocessing pipeline described before.

### J.3.5 TUEP.

The TUEP[16] is one of the datasets in The Temple University Hospital (TUH) Electroencephalography (EEG) Corpus, which is the world's largest open-source EEG corpus. Researchers can access this dataset by submitting a request via an application form to the authors. This dataset is a subset of TUEG and contains data from 100 subjects with epilepsy and 100 subjects without epilepsy, as determined by a certified neurologist.

Each subject in this dataset may have one or more trials, and some trials differ in the number of channels and sampling rates. To ensure consistency, we first select only those subjects whose trials all contain the 19 standard channels: Fp1, Fp2, F7, F3, Fz, F4, F8, T3, C3, Cz, C4, T4, T5, P3, Pz, P4, T6, O1, and O2. A total of 179 subjects meet this requirement. However, because the recording dates of different trials can be widely separated, sometimes by several years, we are concerned that neurological condition shifts may occur, potentially undermining subject-level consistency. Therefore, in this study, we treat each trial as an independent subject, replacing the subject ID with the trial ID for self-supervised contrastive learning. All other preprocessing steps follow the same pipeline described previously.

---

[16]https://isip.piconepress.com/projects/nedc/html/tuh_eeg/

