# OpenReview forum: "LEAD: Large Foundation Model for EEG-Based Alzheimer’s Disease Detection"
_ICLR.cc/2026/Conference — Submitted to ICLR 2026_

### Official Review · Reviewer_qMSz · 2025-10-28

**Soundness:** 2
**Presentation:** 3
**Contribution:** 2
**Rating:** 2
**Confidence:** 3

**Summary:**

This paper introduces LEAD, a large-scale foundation model for EEG-based Alzheimer's Disease (AD) detection. The authors address two primary challenges in the field: the lack of large-scale EEG-AD datasets and the common focus on sample-level detection rather than the more clinically meaningful subject-level detection.

**Strengths:**

1. The proposed methodological additions are directly target the subject-level problem.
2. The paper convincingly shows that its domain-relevant pre-training strategy (using ~730 hours of data) can outperform foundation models pre-trained on much larger, more general EEG corpora (LaBraM at ~2,000 hours, CBraMod at ~9,000 hours). This highlights the importance of data selection for pre-training.

**Weaknesses:**

1. The paper frames itself as the 'first foundation model for EEG-based AD detection'. This premise is arguably a contradiction in terms. Foundation models (such as the baselines LaBraM and CBraMod ) are intended to learn general, robust, and task-agnostic representations from massive, broad data. Specific tasks, like AD detection, are meant to be downstream applications solved by fine-tuning these general models. The authors' approach, curating a large but highly specific pre-training corpus of 'domain-relevant' neurological disorders, is a form of specialized transfer learning. It does not constitute the creation of a new 'foundation model' in the accepted sense. It is, rather, a highly effective domain-specific model, and this framing overstates the generality of its contribution.

2. The core technical contribution beyond the (non-trivial) data engineering appears to be incremental. The proposed model is an adaptation of a spatio-temporal transformer, and the key novel components (subject-level loss, index shuffling) are application-specific engineering solutions.

3. More concerningly, the value of the architectural changes is directly contradicted by the paper's own ablation study.
* The "Vanilla Backbone" alone achieves a strong subject-level F1 score of 73.11%. The full LEAD pipeline, with all modules, achieves 75.77%, a marginal improvement of only ~2.66%.
* Adding "Half-Overlapping" segmentation causes performance to drop from 73.11% to 72.12%. Later, adding "Multi-Scale Segmentation" also causes a performance drop from 75.49% to 74.96%, which suggests the architectural components are, at best, offering a negligible and inconsistent benefit. The vast majority of the gain  comes from "+Sample/Subject-Level Pre-training", reinforcing that the contribution is in the pre-training data strategy, not the LEAD architecture itself.

4.  For a tool aimed at Alzheimer's Disease, the most vital clinical task is early detection, which requires distinguishing MCI from HC and AD. The model's performance on this task is weak.
* On the multi-class datasets that include MCI, the subject-level F1 scores are low (~64.3% and ~59.6%, respectively).
* On the largest dataset, CAUEEG (1,379 subjects), the LEAD model's subject-level F1 score (59.61%) is statistically indistinct from, several baselines, such as LaBraM (58.92%).

5. The abstract states the model achieves a "Sensitivity of 90.91%" on the LOSO benchmark , but Table 9 and Appendix F.2 refer to this 90.91% value as the "Subject-level Accuracy". Could the authors please clarify which metric this value represents?

**Questions:**

Please check above. I will reconsider my assessment after checking authors' response.

---

> ### Author Response · Authors · 2025-11-21
> **Response for Question 1**
>
> Thank you for your thoughtful review and for your willingness to reconsider the assessment! We provide detailed responses to each of your questions below. Regardless of your final decision, we truly appreciate the opportunity to clarify our contributions and address your concerns. We are happy to further clarify any points and would welcome any additional questions you may have.
>
> ---
>
> **Q1**: Regarding the use of the term “foundation model,” which might not be suitable.
>
> **A1**: We appreciate the reviewer’s comment and agree that “foundation model” may not perfectly describe a system dedicated specifically to EEG-based AD detection. Below we clarify our motivation and reasons for using this term.
>
> 1) **Extreme heterogeneity of EEG data motivates a unifying large-scale model.** Unlike computer vision and natural language processing, EEG data suffer from strong heterogeneity: recordings are collected under different environments, hardware devices, electrode montages, acquisition protocols, and from subjects with very different physiological characteristics. These factors introduce substantial domain shifts (e.g., device differences, montage mismatch, varying Signal-to-Noise Ratio (SNR) caused by diverse preprocessing pipelines). As a result, zero-shot or few-shot transfer across datasets is extremely challenging, even when the downstream datasets are all for AD detection. Our motivation is precisely to train a model on a sufficiently diverse collection of AD-related datasets so that the resulting representation is robust when transferred to previously unseen datasets, where subject numbers are often limited due to the difficulty of collecting AD patient data. Since many downstream AD datasets are too small to train a model from scratch, a large pre-trained model that aims to generalize across heterogeneous EEG domains seemed to align, conceptually, with the idea of a “foundation model.” Similar application-specific foundation models have also been explored in the EEG domain, such as those for motor imagery[1].
>
>
>
> 2) **Our model is inherently a broader “brain disorder foundation model,” not limited to AD.**
> As detailed in the paper, beyond AD subjects, our pre-training corpus includes patients with a wide range of neurological and psychiatric disorders, such as Parkinson’s disease, multiple sclerosis, schizophrenia, depression, ADHD, and many comorbid combinations. In total, our pre-training data cover over 50 types of disorder-related conditions. Resting-state EEG collected for these disorders typically follows similar acquisition paradigms, enabling the model to learn more generalizable neural representations. Therefore, although our current work focuses on AD detection, the pre-trained model naturally extends to other brain disorder classification tasks with minimal adaptation. In this sense, the model functions as a “brain disorder foundation model,” with AD detection being one of its downstream applications. We could easily add additional tasks (e.g., Parkinson’s or ADHD detection) in later versions to further generalize it, but such extensions are beyond the scope of this submission aims for AD detection.
>
> Given the reviewer’s concern, we are open to revising the title. One possible alternative title could be: "A Brain Disorder Foundation Model and Its Application to EEG-based Alzheimer’s Disease Detection." Do you have any suggestions regarding the most appropriate naming?

---

> ### Author Response · Authors · 2025-11-21
> **Response for Question 2**
>
> **Q2**: Regarding the novelty of this paper.
>
> **A2**: We thank the reviewer for seeing our effect on the AD datasets curating. Below we clarify the novelty and contributions of our work.
>
> 1) Our work is application-oriented, and we intentionally prioritize effectiveness over unnecessary architectural complexity. Our goal is to build a practical and robust model for EEG-based AD detection. To ensure this, we followed Occam's razor rule by graduately add designs: we systematically tried many theoretically promising modules but only kept those that consistently improve performance. During the development process, we conducted extensive trials, including: (1) **Frequency-aware embeddings:** phase, amplitude, and PSD embeddings added to temporal patches. (2) **Alternative channel embedding strategies:** 2D relative positional encoding, 3D electrode coordinates. (3) **Mitigating subject-level inconsistency:** global prototype consistency via momentum updates. (4) **Backbone variations:** single-channel patching, replacing FFN with mixture-of-experts layers. None of these tried benefit the final performance, while increased computational cost. Although some of these components appear in other EEG foundation model papers, none of them yielded meaningful gains, or even degraded performance, for EEG-based AD detection in our setting. Conversely, the **subject-level cross-entropy loss combined with group index shuffling** proved to be a simple, intuitive, and highly effective way to address the crucial issue of sample-level vs. subject-level inconsistency. Without explicitly modeling subject identity, improvements at the sample level often do not translate to subject-level performance due to overfitting to certain subjects and domain shifts between them. We simple but useful modules in real-world AD detection scenarios are more valuable than fancy but ineffective components merely for novelty in architecture design.
>
>
> 2) Our second contribution is demonstrating the importance of domain-specific pre-training resources and strategies. As reported in the paper, our model—pre-trained on ~700 hours of EEG, where achieves comparable or superior performance to EEG foundation models trained on 10× larger data resources. This indicates two important insights for future research: (1) **Domain-relevant data matter far more than scale alone.** For AD detection, adding more AD subjects during pre-training is substantially more beneficial than adding larger amounts of domain-irrelevant EEG. Similarly, incorporating other brain disorders collected under similar paradigms (e.g., Parkinson’s disease, depression, ADHD) improves the representational space in ways that help AD detection. (2) **Subject-level contrastive learning is crucial and underexplored in large-scale EEG pre-training.** Existing foundation models usually rely on masked reconstruction. However, disease-specific features, especially subtle ones relevant to early-stage AD, are often embedded at the subject level, not the sample level.
> Subject-level contrastive learning is the only self-supervised learning strategy that can capture such features without labels, and, to our knowledge, our work is the first to demonstrate its effectiveness **at large scale**. Importantly, some MCI patients or cognitively normal older adults may already exhibit prodromal AD characteristics before formal diagnosis. Subject-level contrastive learning helps capture these latent features during pre-training, providing an additional advantage for downstream AD detection.

---

> ### Author Response · Authors · 2025-11-21
> **Response for Question 3**
>
> **Q3**: The value of the architectural changes is directly contradicted by the paper's own ablation study.
>
> **A3**: Thank you for raising this concern. We believe the confusion mainly arises from the inherent mismatch between sample-level and subject-level performance in subject-independent EEG disease detection. We clarify this below.
>
>
> 1) **Improvements at the sample level do not necessarily translate into subject-level gains.** As discussed in prior work[4], in subject-level disease detection, increasing sample-level accuracy does not guarantee improved subject-level accuracy. This happens because subject-level evaluation relies on majority voting, which does not participate in back-propagation. A model naturally prioritizes “easy” samples during training, reducing sample-level loss while potentially harming subject-level outcomes. This phenomenon is also clearly reflected in Table 5 and Table 6 of our paper. To illustrate with a simple example: Consider 3 subjects, each with 100 samples. Initial sample-level accuracies: 60%, 80%, 80% (avg: 73%). All three subjects are correctly classified after voting (**100% subject-level accuracy now**). After one epoch: Sample-level accuracies become 40%, 95%, 95% (avg: 76.7%). Sample-level performance improved on average, yet subject 1 is now misclassified (**subject-level accuracy drops to 66.7%**). This is exactly why certain modules in our ablation (Table 5), such as half-overlapping, multi-scale segmentation, and pre-training, consistently improve sample-level results (from 57.95% → 67.87%). This around 10% improvement is huge compared with existing papers on AD detection, which report only sample-level classification. However, these gains alone do not fix the sample/subject inconsistency problem inherent to EEG. This motivates the importance of our last two modules: Subject-level cross-entropy, and Group index shuffling. These components explicitly encourage the model to treat each subject equally rather than each sample, thereby reducing subject-level inconsistency and improving subject-level robustness, as shown in Table 5.
>
>
> 2) **Subject-level performance improves in a threshold-like, non-linear manner.** According to our experimental observation, subject-level performance is not a smooth function of sample accuracy. Certain subjects might sit near the decision boundary (e.g., subjects 23, 25, 59, and 70 in Tables 9 and 10, with ~46%–50% sample-level accuracy). These “boundary subjects” often flip from misclassified to correctly classified only after the model reaches a certain competency threshold. Thus, sample-level improvements (even ≈10%) may still be insufficient to push borderline subjects over this threshold during subject-independent evaluation, resulting in seemingly flat subject-level performance. However, when we increase the number of training subjects, this threshold is more easily surpassed. This is exactly what we observe when comparing cross-validated subject-independent(CVSI) evaluation and leave-one-subject-out(LOSO): CVSI (≈60% training subjects) -> ~75% subject-level accuracy vs. LOSO (≈99% training subjects) -> ~91% subject-level accuracy. The large jump in subject-level accuracy demonstrates that the architecture is indeed effective. However, subject-level gains only emerge when enough subjects are included in the training pool to cross that performance threshold, which could be achieved by collecting more EEG data from different subjects under the same pipeline in a practical scenario.

---

> ### Author Response · Authors · 2025-11-21
> **Response for Question 4 and References**
>
> **Q4**: The model's performance is weak for a tool for Alzheimer's Disease early detection.
>
> **A4**: Here is our response to your concern.
>
> 1) We appreciate the reviewer’s concern and agree that early-stage AD detection critically for distinguishing MCI from HC and AD for early detection. More specfically, the model need to distinguish stable MCI (sMCI) and progress MCI (pMCI)[5][6] from HC and AD. However, none of the datasets used in our work include longitudinal follow-up for each subject, meaning all MCI subjects are labeled simply as “MCI.” This inevitably mixes pMCI and sMCI together, and some pMCI patients may already exhibit AD-like EEG patterns before clinical conversion. Without longitudinal labels, a model cannot reliably separate these subgroups, and this label uncertainty inherently reduces performance. An ideal early-detection system would classify (AD + pMCI) vs. HC vs. sMCI vs. other brain disorders, but the required longitudinal data are not publicly available. Our contribution is to provide a robust training pipeline and a large pre-trained model that researchers with private longitudinal datasets can readily adapt for genuine early-stage prediction.
>
> 2) The performance of EEG-based AD detection is relatively weak if compared with other methods such as Neuroimaging and CSF-based methods, as we discussed in our introduction part in Table 1. EEG is not meant to replace these clinical gold standards; instead, its value lies in being low-cost, safe, non-invasive, and potentially deployable in wearable or BCI-based monitoring systems. Its role is more similar to consumer health technologies, such as smartwatch PPG/ECG-based blood pressure monitors or non-invasive NIR glucose monitors, which cannot replace hospital-grade diagnostics but are still highly useful because they enable frequent, accessible self-monitoring and early risk identification. We envision EEG-based AD detection playing a similar role, which offering routine brain-health assessments and encouraging timely clinical evaluation when abnormal patterns are detected.
>
> 3) Finally, although our performance on the CAUEEG dataset improves only slightly over LaBraM, our model achieves substantially better performance on the other three datasets, surpassing LaBraM by 2.12%, 4.46%, and 4.11%. This is particularly noteworthy given that LaBraM and CBraMod were trained on 2000+ hours and 9000+ hours of EEG respectively, whereas our model uses only ~700 hours of pre-training data. These results highlight the effectiveness of our architectural design, subject-level regularization, and domain-specific pre-training strategy, demonstrating that our approach is competitive and efficient even under the inherent limitations of EEG-based AD detection.
>
> ---
>
> **Q5**: Typo in abstract
>
> **A5**: We thank the reviewer for pointing out this typo in the abstract. The 90.91% is the subject-level accuracy, not sensitivity. In the LOSO evaluation for AD detection, only one subject is left, and all samples belong to that subject has the same label, which means only one class in the test set here. The sensitivity cannot be computed in this case.
>
>
>
>
> ---
> ## **References**
>
> [1] Mirepnet: A pipeline and foundation model for eeg-based motor imagery classification
>
> [2] Medformer: A Multi-Granularity Patching Transformer for Medical Time-Series Classification
>
> [3] CSBrain: A Cross-scale Spatiotemporal Brain Foundation Model for EEG Decoding
>
> [4] ADformer: A Multi-Granularity Spatial-Temporal Transformer for EEG-Based Alzheimer Detection
>
> [5] Sustainable method for Alzheimer dementia prediction in mild cognitive impairment: Electroencephalographic connectivity and
> graph theory combined with apolipoprotein E
>
> [6] Classification and epidemiology of MCI

---

> ### Comment · Reviewer_qMSz · 2025-11-22
>
> I thank the authors for their comprehensive response and for clarifying the typo in the abstract. I appreciate the effort put into the rebuttal and the additional explanations regarding the architectural choices and data limitations.
>
> In recognition of the authors' effort and clarification, **I am raising my score to 4**.
>
> However, my primary concerns regarding the paper's positioning and contribution remain unresolved. Specifically:
>
> ---
>
>
>
> In the Response to Q2, the authors state:
>
> > "Our work is application-oriented, and we intentionally prioritize effectiveness over unnecessary architectural complexity."
>
> While I accept that an application-oriented paper does not strictly require novel algorithmic complexity, the standard for such papers shifts to **demonstrable clinical utility**. As noted in my review and acknowledged by the authors in Response to Q4, the model's performance on early-stage detection (distinguishing MCI from HC and AD) is weak due to the lack of longitudinal labels. For an application-oriented AD detection tool, the ability to detect MCI is arguably the most critical function. A tool that performs well on differentiating clear AD from HC but fails on the subtle early cases has limited real-world clinical application.
>
>
> The paper falls into a difficult middle ground:
>
> - The architectural contribution is incremental (justified by being "application-oriented"), yet the application performance, especially for the most clinically relevant scenario, is not sufficient to be considered a breakthrough solution.
>
> ---
>
>
>
> Even with the proposed title change to *"Brain Disorder Foundation Model"* in Response to Q1, the evaluation remains too narrow.
>
> A foundation model is defined by its ability to generalize to different downstream tasks. Since the paper primarily evaluates AD detection and struggles with the more challenging MCI cases within that domain, the claim of being a "Foundation Model" is not supported by the empirical evidence.
>
> The model currently behaves more like a specialized pre-trained model for AD classification, rather than a generalizable foundation model.
>
> ---
>
>
>
> While I have read the explanation regarding sample-level vs. subject-level mismatch in Response to Q3, the empirical results still imply that:
>
> - Most performance gains stem from **data engineering** (large pre-training corpus)
> - Rather than from the **architectural modules** (group shuffling, subject-level loss)
>
> The data curation is commendable, but the methodological contribution appears marginal.
>
> ---

---

> ### Author Response · Authors · 2025-11-23
> **Response for score raising**
>
> We thank the reviewer for raising the score on our paper and are happy that we addressed most of your concerns! Regarding architectural novelty, we will continue to evolve our method by designing more useful modules to enhance EEG-based AD detection in future work. Regarding performance on MCI classification, we think this might still require years of effort in this domain, including adding more MCI subjects to both pre-training and downstream training corpora and collecting longitudinal data for early-stage AD detection. But we believe our training pipeline and released pre-trained weights will still benefit the researchers in the EEG-AD domain who have their own private longitudinal data. Finally, thank you again for your thoughtful review and suggestions on our paper!

---

### Official Review · Reviewer_Dwx4 · 2025-11-01

**Soundness:** 2
**Presentation:** 3
**Contribution:** 2
**Rating:** 4
**Confidence:** 4

**Summary:**

This paper proposes LEAD, a large-scale, subject-regularized spatio-temporal transformer for EEG-based Alzheimer’s disease detection. It achieves state-of-the-art subject-level performance across four AD datasets.

**Strengths:**

S1. The authors curate the largest unified EEG–AD dataset to date, comprising 2,255 subjects from heterogeneous sources, significantly expanding the available data scale for clinical modeling.

S2. They introduce a subject-regularized spatio-temporal transformer with group shuffling and a subject-level loss to directly optimize subject-level consistency.

S3. The proposed domain-relevant contrastive pretraining effectively enhances representation learning and downstream AD detection performance.

**Weaknesses:**

W1. Signal frequency and channel unification may cause information loss. To handle heterogeneous datasets, all recordings were resampled to 128 Hz and mapped to 19 channels. Although this ensures format consistency, it inevitably sacrifices high-frequency components and temporal resolution.

W2. The COGNISION dataset (7 channels) was spatially interpolated to 19 channels, which achieves formal consistency but weakens the neurophysiological grounding. In addition, Figure 5 and the channel-importance visualization section do not specify their derivation or correspondence to AD-related regions, making the interpretation resemble a simple visual description rather than a physiologically meaningful analysis.

W3. The model processes EEG in the time domain, no frequency-specific representations or analyses are provided. It remains unclear whether the learned patterns align with known AD-related spectral alterations.

W4. The contrastive learning component shows limited methodological innovation. The subject-level contrast appears to simply cluster samples from the same subject rather than capture disease-specific dynamics, and the sample-level contrast largely reuses existing SimCLR/MoCo augmentations.

W5. A few citation styles appear incorrect at sentence endings. The authors may consider using the \citep{} format.

**Questions:**

Q1. Given the variability in diagnostic criteria and MMSE thresholds for AD/MCI/HC among institutions, how did the authors ensure consistent labeling across datasets such as AD-AUDITORY (https://openneuro.org/datasets/ds005048/versions/1.0.0) and ADFTD (https://openneuro.org/datasets/ds004504/versions/1.0.8)? Were any normalization or re-annotation steps applied?

Q2. What is the rationale for fixing the configuration at 128 Hz and 19 channels? Was this choice made for engineering convenience, or was it supported by empirical analysis showing optimal performance?

Q3. Since LEAD’s pretraining remains disease-oriented rather than task-agnostic, in what sense should it be regarded as a foundation model rather than a cross-disease mixed pretraining framework?

---

> ### Author Response · Authors · 2025-11-21
> **Response for Question 1-3**
>
> Thank you for your thoughtful review! We provide detailed responses to each of your questions below. Regardless of your final decision, we truly appreciate the opportunity to clarify our contributions and address your concerns. We are happy to further clarify any points and would welcome any additional questions you may have.
>
>
> ---
>
>
>
> **Q1**: Signal frequency and channel unification may cause information loss. What is the rationale for fixing the configuration at 128 Hz and 19 channels? Was this choice made for engineering convenience, or was it supported by empirical analysis showing optimal performance?
>
> **A1**:
>
> 1) Regarding sampling frequency, our choice of 128 Hz is not based on engineering convenience but on empirical observation and common EEG preprocessing practice. Many raw EEG datasets are recorded at higher rates (e.g., 500 Hz or 1000 Hz), and it is standard to downsample them to 128–256 Hz because this range fully preserves the physiologically relevant EEG spectrum (0.5–45 Hz) while suppressing high-frequency noise. In our prior supervised learning experiments, we directly compared 128 Hz, 200 Hz, and 256 Hz using identical architectures. The results showed no statistically significant differences in downstream AD detection performance. Therefore, we adopted 128 Hz simply because it is the most computationally efficient setting with no evidence of accuracy degradation.
>
> 2) Regarding the unification to a 19-channel 10–20 montage, we initially explored a more flexible training strategy using padding and masked attention, allowing pre-training datasets to retain their native montages. However, preliminary experiments on several pre-training datasets (not all) showed that this approach did not provide improvements in downstream linear probing performance, while significantly increasing computational cost with more memory and longer training convergence. As an alternative, we follow the same design choices used in recent EEG foundation models such as CBraMod[1] and CSBrain[2], which also use common 19-channel 10-20 montage for efficient large-scale pre-training.
>
>
> ---
>
>
> **Q2**: The interpolating of Cognision from 7 channels to 19 channels will weaken the neurophysiological grounding.  Figure-5 visualization does not specify AD-related regions.
>
> **A2**:
>
> 1) The interpolation from 7 to 19 channels does not alter or overwrite the original Cognision signals. Instead, the additional 12 channels are generated through spherical interpolation using the MNE toolbox, which computes linear combinations of the existing 7 channels. Thus, the raw physiological information remains intact, and the interpolation simply provides a consistent spatial layout for the unified model.
>
> 2) Regarding Figure 5, we agree that it does not correspond directly to established AD-related brain regions. However, this is expected. Identifying precise neural sources from scalp EEG is inherently difficult because scalp electrodes do not represent isolated cortical locations but rather linear mixtures of distributed brain activity, which is the classical EEG inverse problem[3]. Therefore, channel importance learned by a deep model should not be interpreted as evidence of specific pathological regions. Our intention for this analysis is to provide practical guidance for future EEG-based AD detection research by indicating which electrode locations carry useful discriminative information in the data, rather than attempting to perform neurophysiological localization. Besides, EEG-based AD detection is not designed to replace gold standard diagnostic methods using neuroimaging or CSF biomarkers; instead, it aims to offer a safe, non-invasive, and low-cost method suitable for wearable-device-based early self-assessment. Similar to emerging non-invasive NIR glucose monitoring research, EEG-based methods are valuable not because they pinpoint exact biological sources, but because they enable broader, more frequent monitoring outside clinical environments.
>
>
> ---
>
>
>
> **Q3**: The model has no frequency-specific representation learning and analysis.
>
>
> **A3**: We did explore frequency-specific representation learning during development. Specifically, we added a PSD-based self-attention branch, and we also experimented with embedding magnitude and phase information, similar to the frequency embedding design in LUNA[4]. However, for AD detection, none of these approaches improved downstream performance. In fact, they often degraded accuracy and slowed training convergence. Since our work is application-oriented, we chose to remove components that add complexity without providing measurable benefits and focused on keeping the model simple, effective, and efficient for real-world AD detection tasks.

---

> ### Author Response · Authors · 2025-11-21
> **Response for Question 4-7**
>
> ---
>
>
> **Q4**: Lack of novelty for contrastive learning components. The subject-level contrast cluster the samples from the same subject rather than capture disease-specific dynamics.
>
> **A4**:
> 1) We would like to clarify that our work is not positioned as a contrastive-learning–method paper but as an application-oriented study focusing on improving EEG-based AD detection. All components in our pipeline, including the choice of domain-relevant pre-training datasets, the transformer backbone, multi-scale segmentation, subject-level cross-entropy, and group index shuffling, are designed for enhancing practical performance. This is demonstrated by the results that our model achieves better or comparable performance than existing EEG foundation models that were trained on 2,000+ or even 9,000+ hours of data, which is much larger than our 700+ hours. Similar to the existing EEG foundation models[1,2,5], we never claim they lack novelty because of using the mask and reconstruction strategy (masked autoencoder) as self-supervised pre-training method.
>
> 2) Disease-related features are inherently embedded within subject-level characteristics. As discussed in Appendix B and D, our assumption is that all samples recorded from the same subject within a short period (e.g. data collected in several days) share a consistent underlying brain state regarding the disease situation[6]. For an AD patient, this means that all their recordings contain AD-related patterns, so treating these samples as positive pairs allows the model to cluster representations that reflect consistent disease-linked properties even without access to labels. This mechanism is especially useful for cases such as progressive MCI or self-reported healthy elders who may already exhibit subtle AD-like features prior to clinical confirmation. In contrast to masked-reconstruction self-supervised learning methods, subject-level contrastive learning is the only strategy that can capture such disease-relevant consistency during pre-training without labels.
>
>
> ---
>
>
> **Q5**:  A few citation styles appear incorrect at sentence endings. The authors may consider using the \citep{} format.
>
> **A5**: We thank the reviewer for correcting our formatting issue. We keep in mind and will revise it in the next version.
>
>
> ---
>
>
> **Q6**:  Given the variability in diagnostic criteria and MMSE thresholds for AD/MCI/HC among institutions, how did the authors ensure consistent labeling across datasets?
>
> **A6**: In short, we did not attempt to unify diagnostic criteria across datasets. Each dataset retains its original labeling scheme. During the self-supervised pre-training stage, we do not use any diagnostic labels. The self-supervised learning using sample- and subject-level contrasting only use, metadata subject ID. For downstream evaluation, each dataset is fine-tuned independently, and we strictly follow its original diagnostic definitions and thresholds when performing classification.
>
>
> ---
>
>
> **Q7**: What is the rationale for fixing the configuration at 128 Hz and 19 channels? Was this choice made for engineering convenience, or was it supported by empirical analysis showing optimal performance?
>
> **Q7**: See our response to **Q2**.

---

> ### Author Response · Authors · 2025-11-21
> **Response for Question 8 and References**
>
> **Q8**: Regarding the use of the term “foundation model,” which might not be suitable.
>
>
> **A8**: We appreciate the reviewer’s comment and agree that “foundation model” may not perfectly describe a system dedicated specifically to EEG-based AD detection. Below we clarify our motivation and reasons for using this term.
>
> 1) **Extreme heterogeneity of EEG data motivates a unifying large-scale model.** Unlike computer vision and natural language processing, EEG data suffer from strong heterogeneity: recordings are collected under different environments, hardware devices, electrode montages, acquisition protocols, and from subjects with very different physiological characteristics. These factors introduce substantial domain shifts (e.g., device differences, montage mismatch, varying Signal-to-Noise Ratio (SNR) caused by diverse preprocessing pipelines). As a result, zero-shot or few-shot transfer across datasets is extremely challenging, even when the downstream datasets are all for AD detection. Our motivation is precisely to train a model on a sufficiently diverse collection of AD-related datasets so that the resulting representation is robust when transferred to previously unseen datasets, where subject numbers are often limited due to the difficulty of collecting AD patient data. Since many downstream AD datasets are too small to train a model from scratch, a large pre-trained model that aims to generalize across heterogeneous EEG domains seemed to align, conceptually, with the idea of a “foundation model.” Similar application-specific foundation models have also been explored in the EEG domain, such as those for motor imagery[1].
>
>
>
> 2) **Our model is inherently a broader “brain disorder foundation model,” not limited to AD.**
> As detailed in the paper, beyond AD subjects, our pre-training corpus includes patients with a wide range of neurological and psychiatric disorders, such as Parkinson’s disease, multiple sclerosis, schizophrenia, depression, ADHD, and many comorbid combinations. In total, our pre-training data cover over 50 types of disorder-related conditions. Resting-state EEG collected for these disorders typically follows similar acquisition paradigms, enabling the model to learn more generalizable neural representations. Therefore, although our current work focuses on AD detection, the pre-trained model naturally extends to other brain disorder classification tasks with minimal adaptation. In this sense, the model functions as a “brain disorder foundation model,” with AD detection being one of its downstream applications. We could easily add additional tasks (e.g., Parkinson’s or ADHD detection) in later versions to further generalize it, but such extensions are beyond the scope of this submission aims for AD detection.
>
> Given the reviewer’s concern, we are open to revising the title. One possible alternative title could be: "A Brain Disorder Foundation Model and Its Application to EEG-based Alzheimer’s Disease Detection." Do you have any suggestions regarding the most appropriate naming?
>
>
>
>
> ---
>
> ## **References**
>
> [1] Cbramod: A criss-cross brain foundation model for eeg decoding
>
> [2] CSBrain: A Cross-scale Spatiotemporal Brain Foundation Model for EEG Decoding
>
> [3] EEG Signal Processing and Feature Extraction
>
> [4] LUNA: Efficient and Topology-Agnostic Foundation Model for EEG Signal Analysis
>
> [5] Large brain model for learning generic representations with tremendous EEG data in BCI
>
> [6] Contrast Everything: A Hierarchical Contrastive Framework for Medical Time-Series

---

> > ### Comment · Reviewer_Dwx4 · 2025-11-25
> >
> > Thanks for the detailed clarifications. I’d like to offer a bit more feedback.
> >
> > Several of your responses mention design choices are based on negative results or unsuccessful attempts. For example:
> >
> > - "we directly compared 128 Hz, 200 Hz, and 256 Hz … no statistically significant differences in downstream AD detection performance"
> > - "preliminary experiments on several pre-training datasets … not provide improvements in downstream linear probing performance …"
> > - "experimented with embedding magnitude and phase information …  none of these approaches improved downstream performance"
> >
> > Including these observations together with brief explanations or illustrative results in the manuscript would actually be valuable for the community. Since the rebuttal phase allows for manuscript revisions and extends page limits, I suggest you incorporate these insights and your responses to other reviewers directly into the main text.
> >
> > Regarding the positioning of the work:
> >
> > - "our work is not positioned as a contrastive-learning–method paper but as an application-oriented study focusing on EEG-based AD detection," and
> > - "the model functions as a 'brain disorder foundation model,' with AD detection being one of its downstream applications."
> >
> > It is hard to reconcile these two claims. In BCI research, "foundation models" usually imply task-agnostic representation learning for a wide variety of downstream tasks. In contrast, AD detection is a much narrower problem setting. In this context, I agree with Reviewer qMSz that the current empirical evidence doesn't quite support the "foundation model" label.
> >
> > Consider more precise terminology, such as Large Pretrained Model/Pretraining Framework, might be better.

---

> > > ### Author Response · Authors · 2025-11-25
> > >
> > > We thank the reviewer for the valuable feedback and suggestions! Some prior experiments were conducted during our model design phase, but the results were not formally saved for five seeds. We will include these failed attempts in the appendix of our next revision, like an ablation study, along with any new findings and changes to our model architecture. Besides, we will discuss with co-authors and reconsider the title and rename it to things like "large pre-trained model for AD detection", as you suggested. We thank you again for your thoughtful review and suggestions on our paper!

---

### Official Review · Reviewer_thwC · 2025-11-01

**Soundness:** 3
**Presentation:** 3
**Contribution:** 2
**Rating:** 6
**Confidence:** 4

**Summary:**

The paper introduces a foundation model trained to process EEG data from pathological individuals, with a focus on Alzheimer’s disease detection downstream tasks at the subject level. A transformer architecture with separate temporal and spatial branches is first pretrained using contrastive objectives at the sample and subject levels. The model is then finetuned to perform Alzheimer’s disease-related classification tasks such as pathology type or stage classification. For this, a corpus of public and private datasets (2255 subjects, 442h of recording) was curated and used to pretrain and evaluate performance of the model. Results suggest that in most cases the proposed model outperforms other pipelines based on handcrafted features, fully supervised architectures, or “generalist” EEG foundation models.

**Strengths:**

Originality
* The paper presents the first EEG foundation model pretrained specifically on pathological data with a focus on Alzheimer’s disease. While there are multiple papers in the literature focused on building EEG foundation models, I am not aware of any that are specifically focused on this subtype of data.
* Similarly, the curation and unified preprocessing of a large corpus of AD-specific datasets appears novel.

Quality
* The proposed method is compared to several relevant baselines (handcrafted features approach, supervised models, and two other “generalist” EEG foundation models), on 4 Alzheimer’s disease datasets from different sources. These comparisons suggest that the proposed model almost always outperforms baselines, which supports the claim of foundation model training.
* The provided ablation studies suggest adding pretraining steps yield the largest performance increase, which supports the use of a pretraining-finetuning pipeline. The ablation study on the pretraining corpus is also informative and supports the use of the curated corpus.

Clarity
* The paper is overall well written and easy to follow, and technical decisions are well supported.

Significance
* Based on the reported strong results on the 4 downstream datasets, this paper is likely to be relevant for the EEG and Alzheimer’s disease prediction community. More generally, this can provide a strong point of comparison for evaluating “generalist” EEG foundation models on pathological EEG detection tasks.

**Weaknesses:**

1. While the baseline approaches are diverse, comparisons with architectures that have been specifically designed and/or tested on AD data in the past appear to be missing. Appendix A lists a few of them, which I believe are not considered in the results of Tables 3 and 7. Most notably, the last model to be listed, ADformer, presents many similarities to the proposed work both in terms of architecture and evaluation protocol. Similarly, it would be interesting to compare performance to existing EEG foundation models that have also tested their models on similar tasks, for instance [1]. See Q1.

[1] Yuan, Zhizhang, et al. "Brainwave: A brain signal foundation model for clinical applications." arXiv preprint arXiv:2402.10251 (2024).

2. The reason for focusing on subject-level evaluation is well explained in the manuscript. However, it seems to me that the proposed approach relies on the same strategy as previous work to yield subject-level predictions, i.e. majority voting from sample-level predictions. Along with the limited performance increase yielded by the subject cross-entropy loss (Table 5), this weakens the listed contribution of building a novel subject-level detection pipeline.

3. The contrastive learning strategy at the core of the paper is taken from a previous paper (Wang et al., 2023), which limits the novelty of this work on the algorithm side. Of note, this reference is only mentioned in the appendix, whereas it seems central enough that it should be cited in the main text.

4. The preprocessing pipeline relies on heavy preprocessing steps, i.e. ICA and rejection of artifactual components, and requires adapting electrode montages to a common 19-channel montage. Given existing foundation models have shown promising results with limited or no preprocessing steps applied (e.g. LaBraM and CBraMod), with flexible montage support, this limits the flexibility of the proposed model in comparison (see Q2). Additionally, filtering the signals below 45 Hz may get rid of relevant information (see Q3).

**Questions:**

1. How does LEAD compare to existing deep learning approaches specifically designed for AD-related classification tasks?
2. Have the authors evaluated the importance of the artifact correction step?
3. Cutting off information in higher frequency bands (above 45 Hz) may be limiting as information in the gamma band has been reported as relevant for AD classification, e.g. in Wang et al. (2017). Why was this choice made?

---

> ### Author Response · Authors · 2025-11-21
> **Response for Question  1-2**
>
> Thank you for your thoughtful feedback on our work！ We appreciate your careful reading and are happy to receive your endorsement, and feel that our paper is easy to follow! We are happy to clarify any points further and would welcome any additional questions you may have. Again, thank you for your comments. Here are the responses to your questions and concerns.
>
> ---
>
> **Q1**: Missing comparison with AD-specific methods and foundation model on simular task
>
> **A1**:
>
> 1) We additionally include comparisons with AD-specific methods, including STEADYNet[1], MNet[2], and ADformer[3]. The detailed results are shown in the newly added table R1. Our method demonstrates a clear performance advantage over all existing AD-specific approaches. Notably, STEADYNet performs close to random chance on the 3-class setting in our subject-independent evaluation, suggesting that previously reported high accuracy in many AD detection papers, typically obtained on small datasets or under subject-dependent evaluation protocols, may not reliably translate to real-world AD detection scenarios.
>
>
>     ### **Table R1: AD-Specific methods**
>     |Subject-Level|Accuracy|Precision|Recall(Sensitivity)|F1 Score|AUROC|AUPRC|
>     |-|-|-|-|-|-|-|
>     |STEADYNet|49.47±10.84|44.28±21.06|45.06±10.31|37.05±15.68|58.96±7.80|42.14±8.29|
>     |MNet|66.32±8.55|70.88±9.23|64.28±5.57|60.65±5.78|73.80±4.65|55.00±6.16|
>     |ADformer|77.89±6.14|81.94±1.39|75.50±6.36|74.55±7.51|82.10±4.55|68.31±6.56|
>     |LEAD|**77.89±5.16**|**82.81±2.42**|**76.72±5.94**|**75.77±5.81**|**82.77±4.28**|**68.73±5.90**|
>
>
>
>
> 2) For the foundation model Brainwave [11], thank you for bringing this excellent work to our attention. We will include it in the related work section in a later revision. Because the authors have not released their code or pre-trained weights, we base our comparison on the results they reported for the ADFTD dataset. In their paper, they refer to the AD-65 subset of ADFTD, which contains 65 AD and healthy control subjects, and report a sample-level AUROC of approximately **75%** (Figure 2). In contrast, we report AD vs. HC Sensitivity and F1 scores for all datasets in Table 8 of the appendix, and provide the complete results in Table R2. Our sample-level AUROC on ADFTD is approximately **89%**. While it would be arbitrary to claim superiority without access to their exact subject splits or evaluation protocol, our performance is at least comparable, and potentially stronger, under equivalent conditions.
>
>
>     ### **Table R2: AD vs. HC Full result on ADFTD**
>     ||Accuracy|Precision|Recall(Sensitivity)|F1 Score|AUROC|AUPRC|
>     |-|-|-|-|-|-|-|
>     |Sample-level|80.50±3.97|80.54±3.94|80.75±4.00|80.46±3.96|**89.01±2.76**|88.45±2.63|
>     |Subject-level|92.86±4.52|92.80±4.61|93.33±4.64|92.78±4.61|93.33±4.64|93.53±5.33|
>
>
> ---
>
> **Q2**: The work relies on previous public paper COMET[4] on contrastive learning framework. This work should be cited in the main text.
>
> **A2**:
>
> We do cite COMET in the main text (line 294). The reason we place the contrastive-learning loss equations in the appendix is precisely because, as the reviewer noted, these components are not claimed as contributions of our work as they were defined by prior research. We include them simply for completeness of the self-supervised pre-training pipeline, similar to how other EEG foundation models using masked-reconstruction strategies. We adopt COMET because it is currently the only self-supervised approach that can capture certain disease-related patterns during pre-training without labels (as discussed in Appendix B and D), which is essential for large-scale EEG disease modeling and early-stage pattern findings.

---

> ### Author Response · Authors · 2025-11-21
> **Response for Question 3**
>
> **Q3**: Lack of novelty on algorithm side, and subject-level cross-entropy loss yield little performance improvement.
>
>
> **A3**:
>
> 1) Our work is application-oriented, and we intentionally prioritize effectiveness over unnecessary architectural complexity. Our goal is to build a practical and robust model for EEG-based AD detection. To ensure this, we followed Occam's razor rule by graduately add designs: we systematically tried many theoretically promising modules but only kept those that consistently improve performance. During the development process, we conducted extensive trials, including: (1) **Frequency-aware embeddings:** phase, amplitude, and PSD embeddings added to temporal patches. (2) **Alternative channel embedding strategies:** 2D relative positional encoding, 3D electrode coordinates. (3) **Mitigating subject-level inconsistency:** global prototype consistency via momentum updates. (4) **Backbone variations:** single-channel patching, replacing FFN with mixture-of-experts layers. None of these tried benefit the final performance, while increased computational cost. Although some of these components appear in other EEG foundation model papers, none of them yielded meaningful gains, or even degraded performance, for EEG-based AD detection in our setting. Conversely, the **subject-level cross-entropy loss combined with group index shuffling** proved to be a simple, intuitive, and highly effective way to address the crucial issue of sample-level vs. subject-level inconsistency. Without explicitly modeling subject identity, improvements at the sample level often do not translate to subject-level performance due to overfitting to certain subjects and domain shifts between them. We simple but useful modules in real-world AD detection scenarios are more valuable than fancy but ineffective components merely for novelty in architecture design.
>
>
> 2) Our second contribution is demonstrating the importance of domain-specific pre-training resources and strategies. As reported in the paper, our model—pre-trained on ~700 hours of EEG, where achieves comparable or superior performance to EEG foundation models trained on 10× larger data resources. This indicates two important insights for future research: (1) **Domain-relevant data matter far more than scale alone.** For AD detection, adding more AD subjects during pre-training is substantially more beneficial than adding larger amounts of domain-irrelevant EEG. Similarly, incorporating other brain disorders collected under similar paradigms (e.g., Parkinson’s disease, depression, ADHD) improves the representational space in ways that help AD detection. (2) **Subject-level contrastive learning is crucial and underexplored in large-scale EEG pre-training.** Existing foundation models usually rely on masked reconstruction. However, disease-specific features, especially subtle ones relevant to early-stage AD, are often embedded at the subject level, not the sample level.
> Subject-level contrastive learning is the only self-supervised learning strategy that can capture such features without labels, and, to our knowledge, our work is the first to demonstrate its effectiveness **at large scale**. Importantly, some MCI patients or cognitively normal older adults may already exhibit prodromal AD characteristics before formal diagnosis. Subject-level contrastive learning helps capture these latent features during pre-training, providing an additional advantage for downstream AD detection.

---

> ### Author Response · Authors · 2025-11-21
> **Response for Question 4 and References**
>
> **Q4**: The preprocessing pipeline relies on heavy preprocessing steps, such as frequency filtering under 45Hz, artifactual rejection, and 19 channel montage setting. These steps limit the flexibility compared with existing EEG foundation models. What is the importane of these steps?
>
> **A4**:
>
> 1) For frequency filtering, the 0.5–45 Hz range is the most widely adopted setting in EEG-based AD and dementia detection[4]. AD-related abnormalities primarily appear in the delta (0.5–4 Hz), theta (4–7 Hz), alpha (8–12 Hz), and beta (13–30 Hz) bands. Gamma (>30 Hz) signals are easily contaminated by EMG, line noise, and other non-neural artifacts. Some studies even use 0.5–30 Hz to fully remove gamma noise. In contrast, EEG foundation models targeting tasks such as seizure detection often keep up to 75 Hz[7] because high-frequency spikes contain diagnostic value in epilepsy. In dementia detection, however, including gamma frequencies >45Hz may artificially introduce spurious correlations between non-EEG features and the label. For example, dementia patients may exhibit abnormal muscle activity that appears in the gamma band, but such patterns are heavily environment- and pipeline-dependent and do not generalize. This is similar to an image classifier “recognizing” cows based on grass backgrounds: the correlation helps accuracy but does not reflect the true target signal.
>
>
> 2) For artifact rejection, our goal is likewise to eliminate spurious correlations rather than to increase accuracy directly. Artifact removal does not necessarily improve performance, but EEG-based disease detection should not rely on muscle activity or eye movements. Such non-neural artifacts are often confounded with disease labels in small clinical datasets. Heavy preprocessing helps force the model to learn disease-related neural patterns rather than shortcuts. In comparison, other EEG foundation models typically include artifacts in pre-training because their downstream tasks, such as TUEV[7], explicitly require classifying artifact events, so retaining artifacts is beneficial for their scenario.
>
>
>
>
> 3) For montage setting, using a unified 19-channel 10–20 montage for pre-training is standard practice in many foundation models, including CBraMod[7] and CSBrain[8]. These works show comparable or even better results than models pre-trained on higher-density montages (e.g., EEGPT[9], LaBraM[10]). We also conducted preliminary experiments on several pre-training datasets using padding and masked attention to handle arbitrary channel layouts. However, this approach brought no improvement in linear-probe performance while significantly increasing training time. Thus, for pre-training we adopt the more efficient unified-montage pipeline. For downstream tasks, the montage is flexible: we additionally experimented with a learnable 1D convolutional channel-mapping module (learnable interpolation), and it even outperformed the MNE-based interpolation used in the current paper. We will include these results in the revised version.
>
>     ### **Table R2: Interpolation Method Comparison**
>     |Subject-Level|Accuracy|Precision|Recall(Sensitivity)|F1 Score|AUROC|AUPRC|
>     |-|-|-|-|-|-|-|
>     |MNE Interpolation|70.81±3.97|71.26±3.97|70.62±4.52|70.30±4.45|70.62±4.52|68.34±4.18|
>     |1D-Conv Interpolation|**71.89±5.30**|**72.59±5.45**|**72.15±5.28**|**71.72±5.25**|**72.15±5.28**|**70.04±4.62**|
>
>
> ---
>
> ## **References**
> [1] STEADYNet: Spatiotemporal EEG analysis for dementia detection using convolutional neural network
>
> [2] A deep learning model for the detection of various dementia and MCI pathologies based on resting-state electroencephalography data: A retrospective multicentre study
>
> [3] ADformer: A Multi-Granularity Spatial-Temporal Transformer for EEG-Based Alzheimer Detection
>
> [4] Contrast Everything: A Hierarchical Contrastive Framework for Medical Time-Series
>
> [5] Machine learning algorithms and statistical approaches for Alzheimer's disease analysis based on resting-state EEG recordings: A systematic review
>
> [6] A convolutional neural network approach for classification of dementia stages based on 2D-spectral representation of EEG recordings
>
> [7] Cbramod: A criss-cross brain foundation model for eeg decoding
>
> [8] CSBrain: A Cross-scale Spatiotemporal Brain Foundation Model for EEG Decoding
>
> [9] EEGPT: Pretrained transformer for universal and reliable representation of eeg signals
>
> [10] Large brain model for learning generic representations with tremendous EEG data in BCI
>
> [11] BrainWave: A Brain Signal Foundation Model for Clinical Applications

---

### Official Review · Reviewer_fDC2 · 2025-11-01

**Soundness:** 2
**Presentation:** 3
**Contribution:** 2
**Rating:** 4
**Confidence:** 4

**Summary:**

The paper introduces LEAD, a large-scale model for EEG-based Alzheimer’s Disease detection that is pre-trained on 3 AD datasets and 9 non-AD datasets by employing both sample- and subject-level contrastive learning, followed by fine-tuning on 60% of subjects from 4 AD datasets with joint sample- and subject-level cross-entropy losses, while the remaining 40% are reserved for validation and evaluation. Their main contribution is integrating a multi-scale segmentation strategy, a subject-regularized spatio–temporal
transformer with both sample- and subject-level cross-entropy losses, and an indices group-shuffling algorithm to reduce differences between sample-level performance  and subject-level detection. The authors fine-tuned and tested the model on four datasets,  mainly report sensitivity and F1-scores and report performance improvements in comparison to 10 baseline results. In addition, the authors conducted an ablation study to evaluate the effectiveness of each domain-relevant pre-training dataset and an ablation study on the ADFTD dataset to assess the contribution of each module.

**Strengths:**

* Performance comparison among different groups on the ADFTD dataset
* The paper provides an ablation study of modules, and an ablation study of pre-training datasets.
* binary and multi-class results are provided
* Each evaluation is repeated with five random seeds reporting the mean and standard deviation
* creating larger, open-source datasets from heterogeneous clinical populations  is important to address  the lack of large-scale EEG datasets and is essential to investigate cross-institutional generalisability.

**Weaknesses:**

* In addition, Leave-One-Subject-Out Analysis is provided, however it is limited to one dataset (ADFTD) and only for the LEAD model.
* The information in the repository does not match that in the document. For example repository: 9AD and 7 non-AD, 5 AD for finetuning; Paper: 7AD, 9 non-AD, 4 AD for finetuning.
* The authors compare the performance of the LEAD model to 10 baselines. Two models are foundation models, however the other models are not designed for AD detection.
* A direct comparison of the results to previously published state-of-the-art results on the four datasets is not provided, nor is any information given about the typical range of classification performance for AD detection.
* For a direct comparison, please provide the sensitivity for 95% specificity, and also other performance metrics, such as accuracy, balanced accuracy, specificity, PR curves, ROC curves, AUCs, etc.
detailed analysis is focused on the ADFTD dataset, no detailed analysis for the other three evaluation sets
* Overall, multi-class classification performances (Sensitivity and F1-score) of LEAD are in the range of 60-77%, and for binary classification in the range of 60-93%. On the benchmark ADFTD dataset, the  model achieves a subject-level sensitivity of 90.91% under the leave-one-subject-out (LOSO) setting.  However, it is unclear how this compares to previous state-of-the-art results.
* The baselines used in this work are mostly designed for other EEG tasks:
  - EEGNet: BCI
  - TST: multivariate time series datasets (heartbeat, Facedetection, etc; no EEG)
  - EEGInception: Motor Imagery BCI
  - EEGConformer: BCI motor imagery and emotion recognition paradigms
  - BIOT: pre-trained on 6 datasets, fine-tuned on abnormal EEG detection
  - Medformer: included AD detection, multi-class performance on ADFTD: 97.50% F1 score
  - MedGNN:  medical time series classification: sample-based F1score ADFD dataset: 98.30%
  - LaBraM: pre-trained foundation model evaluated on various EEG detection tasks, such as including abnormal detection, event type  classification, emotion recognition, and gait prediction
  - CBraMod: pre-trained foundation model for various  BCI tasks

* A statistical evaluation is missing.
* It remains unclear what clinical improvements this approach provides, especially compared to previously introduced approaches.

**Questions:**

* Why are the ADFTD, CNBPM, Cognision, CAUEEG chosen as final dataset for fine-tuning and evaluation and not AD-Auditory, BrainLat and P-ADIC?
* What are the criteria for selecting the baseline? Why did you not use DNN designed for AD detection, such as SteadyNet, MNet or ADformer?
* Are the baselines trained using the subject-level cross-entropy loss and indices group-shuffling?
* To what extent is the imbalance of the dataset taken into account in relation to the loss function?
* How does the performance of LEAD compare to other models on datasets that have not been pre-trained on multiple datasets?
* Combining multiple EEG datasets, there are several key factors that have to be considered, including data heterogeneity and acquisition variability, institutional biases (distribution mismatch), demographic disparities (dataset imbalance, physical characteristics), data governance and handling. What kind of methodological and technical solutions (e.g. data harmonisation) are applied to address these challenges?

Suggestions:

* Include a discussion of the challenges faced when combining multiple EEG datasets and how they were addressed in this work.
* For a more robust comparison, please include baselines that have been applied for AD detection, such as SteadyNet, ADformer or MNet, as well as a detailed comparison to previous published results on these four datasets.

---

> ### Author Response · Authors · 2025-11-21
> **Response for Question 1-6**
>
> Thank you for your thoughtful review! We provide detailed responses to each of your questions below. Regardless of your final decision, we truly appreciate the opportunity to clarify our contributions and address your concerns. We are happy to further clarify any points and would welcome any additional questions you may have.
>
> ---
>
> **Q1**: Only use  Leave-One-Subject-Out (LOSO) for one dataset.
>
>
> **A1**:
>
> 1) **CVSI is a substantially more challenging evaluation protocol than LOSO.** Cross-validated subject-independent (CVSI) evaluation trains on only about 60% of the subjects, whereas LOSO leaves only one subject for testing, effectively using ~99% of the subjects for training. This difference is clearly reflected in our ADFTD results: LOSO achieves a subject-level F1 score of 91%, while CVSI achieves only 76%. Thus, CVSI provides a stricter and more realistic assessment of model generalization.
>
> 2) **LOSO is impractical for large datasets.** For datasets such as CAUEEG with thousands of subjects, LOSO would require thousands of independent training runs, which is computationally prohibitive and unnecessary. CVSI already produces balanced subject splits and yields a reliable estimate of generalization for large-scale datasets.
>
> 3) **We use LOSO only where it enables meaningful subject-wise analysis.** We adopt LOSO for ADFTD because this dataset includes detailed public demographic information, allowing us to provide subject-level analyses and interpretability grounded in demographics.
> For all other datasets, CVSI suffices for fair comparison with prior work and for demonstrating the effectiveness of our model design.
>
> ---
>
> **Q2**: The information in paper is mismatch with the repository
>
>
> **A2**: The code currently available in the repository is an earlier demonstration version. Because this work is part of a patent application, we will release the updated and finalized code after the patent submission is completed. In the latest version used for this paper, we removed three smaller datasets, added two additional brain-disorder datasets, and adjusted which datasets are used for pre-training and fine-tuning. These updates will be fully reflected when the final code release becomes available.
>
>
> ---
>
> **Q3 & Q4 & Q5**: Author need to provide direction comparision to previous published SOTA method on these four datasets. And method designed for EEG-based AD detection need to compare with.Besides, details analysis is need for other three datasets
>
>
> **A3 & A4 & A5**:
>
> 1) On the ADFTD dataset, we have compared with Medformer[1] and MedGNN[2], two SOTA methods on ADFTD under the same evaluation setup and data split. We newly added three EEG-AD detection SOTA methods, STEADYNet[3], MNet[4], and ADformer[5], for comparison as you suggested. The detailed results are shown in the newly added table R1. Our method demonstrates a clear performance advantage over all existing AD-specific approaches. Notably, STEADYNet performs close to random chance on the 3-class setting in our subject-independent evaluation, suggesting that previously reported high accuracy in many AD detection papers, typically obtained on small datasets or under subject-dependent evaluation protocols, may not reliably translate to real-world AD detection scenarios.
>
>
>     ### **Table R1: AD-Specific methods**
>     |Subject-Level|Accuracy|Precision|Recall(Sensitivity)|F1 Score|AUROC|AUPRC|
>     |-|-|-|-|-|-|-|
>     |STEADYNet|49.47±10.84|44.28±21.06|45.06±10.31|37.05±15.68|58.96±7.80|42.14±8.29|
>     |MNet|66.32±8.55|70.88±9.23|64.28±5.57|60.65±5.78|73.80±4.65|55.00±6.16|
>     |ADformer|77.89±6.14|81.94±1.39|75.50±6.36|74.55±7.51|82.10±4.55|68.31±6.56|
>     |LEAD|**77.89±5.16**|**82.81±2.42**|**76.72±5.94**|**75.77±5.81**|**82.77±4.28**|**68.73±5.90**|
>
>
>
> 2) The other three datasets do not provide publicly available demographic information. In addition, as discussed in A1, applying LOSO to CAUEEG is impractical due to the extremely large number of subjects.
>
>
> ---
>
> **Q6**: Some compared methods have very high performance reported in their paper.
>
>
> **A6**: The results you cited for Medformer and MedGNN correspond to a subject-dependent evaluation setup. As stated in their original paper, the reported accuracies of 97.50% and 98.30% on the ADFTD dataset were obtained under subject-dependent settings for quick testing only. The meaningful real-world meaningful results under subject-independent evaluation on ADFTD are 50.65% and 55.00% sample-level F1 scores for three-class classification, as reported in their paper. However, since their subject splits were fixed for each run, these results are not directly comparable to ours, as we use cross-validated subject-independent evaluation, which is a more robust and reliable assessment protocol. We achieve 67.41% sample-level F1 score under this protocol, compared with 57.54% and 59.98% for Medformer and MedGNN, respectively.

---

> ### Author Response · Authors · 2025-11-21
> **Response for Question 7-12**
>
> **Q7**: It remains unclear what clinical improvements this approach provides, especially compared to previously introduced approaches.
>
>
> **A7**: Compared with existing supervised-learning approaches, our method consistently demonstrates clear performance improvements across all four datasets, including the three additional AD-specific baselines newly added in A3, A4, and A5. In terms of foundation models, our approach achieves superior or comparable performance while using only ~700 hours of pre-training data, whereas existing EEG foundation models rely on 2,000+ hours or even 9,000+ hours of training resources. These results highlight the effectiveness of our architectural design, subject-level regularization, and domain-specific pre-training strategy, demonstrating that our approach is competitive and efficient even under the inherent limitations of EEG-based AD detection.
>
> ---
>
> **Q8**: Why are the ADFTD, CNBPM, Cognision, CAUEEG chosen as final dataset for fine-tuning and evaluation and not AD-Auditory, BrainLat and P-ADIC?
>
>
> **A8**: As shown in Table 2 dataset statistics of our paper, both BrainLat and P-ADIC contain a large number of subjects with disorders other than Alzheimer’s disease or healthy controls, including multiple sclerosis, depression, Parkinson’s disease, and frontotemporal dementia. We aim to fully utilize these heterogeneous brain-disorder subjects as part of our pre-training resources rather than mixing them into downstream AD-specific evaluation, which would compromise the interpretability and clinical clarity of the fine-tuning results. Besides, regarding AD-Auditory, its total number of usable subjects is extremely limited (35 in total, with two subjects missing labels). Such a small subject size does not provide a reliable or generalizable downstream evaluation for disease detection. However, it is still valuable as part of the pre-training corpus. For these reasons, ADFTD, CNBPM, Cognision, and CAUEEG were selected as the final fine-tuning and evaluation datasets, as they provide sufficient subject counts and clear AD-related diagnostic labels required for a meaningful subject-independent evaluation.
>
> ---
>
> **Q9**: Why did you not use DNN designed for AD detection, such as SteadyNet, MNet or ADformer?
>
>
> **A9**: See our results and analysis in **A3 & A4 & A5**.
>
> ---
>
> **Q10**: Are the baselines trained using the subject-level cross-entropy loss and indices group-shuffling?
>
>
> **A10**: No. These components are newly proposed in our method and are not used in any of the baseline models. They were introduced after extensive engineering exploration, and we include them as part of our final training pipeline because they consistently improved robustness and subject-level performance.
>
>
> ---
>
> **Q11**: To what extent is the imbalance of the dataset taken into account in relation to the loss function?
>
>
> **A11**: Our loss function does not explicitly account for dataset imbalance. The subject-level cross-entropy loss is computed by the group_size number of samples per subject in a batch, and the group_size is comtrolled by index shuffling algorithm (See our Algorithm 1 and 2 in paper).  These two combinations enforce a minimum number of samples from each subject within a batch. This mechanism ensures that subjects contribute more evenly during training, independent of the original distribution of samples across subjects or datasets, but it does not incorporate class-imbalance weighting.
>
>
> ---
>
>
> **Q12**: How does the performance of LEAD compare to other models on datasets that have not been pre-trained on multiple datasets?
>
>
> **A12**: The comparisons are detailed in our response for **A3, A4, and A5**, as well as in Table 3 of the main paper. Across all datasets, LEAD consistently outperforms prior supervised baselines and existing EEG foundation models, even though those models were often trained on substantially larger pre-training corpora. These results demonstrate that our design, including the domain-relevant pre-training strategy, multi-scale segmentation, and subject-level optimization—yields stronger generalization even without relying on extremely large heterogeneous pre-training datasets.

---

> ### Author Response · Authors · 2025-11-21
> **Response for Question 13 and References**
>
> **Q13**: What kind of methodological and technical solutions (e.g. data harmonisation) are applied to address challenges of data heterogeneity?
>
>
> **A13**: As described in Section 2.1 of our original paper, we perform a unified preprocessing pipeline to harmonize all datasets before pre-training. This includes consistent artifact removal, standardized sampling rates, unified frequency filtering, identical segmentation rules, channel montage selection, and normalization procedures. These steps ensure that EEG signals from different sources are aligned under the same preprocessing configuration, reducing domain shifts caused by heterogeneous acquisition protocols. In addition, as discussed in our response to reviewer **thwC** in **A4**, the downstream montage is now flexible as long as applying a learnable channel-mapping convolution module. This allows the model to handle datasets with arbitrary channel configurations after pre-training, further mitigating heterogeneity issues across datasets.
>
> ---
>
>
> ## **References**
>
>
> [1] Medformer: A multi-granularity patching transformer for medical time-series classification
>
> [2] Towards multi-resolution spatiotemporal graph learning for medical time series classification
>
> [3] STEADYNet: Spatiotemporal EEG analysis for dementia detection using convolutional neural network
>
> [4] A deep learning model for the detection of various dementia and MCI pathologies based on resting-state electroencephalography data: A retrospective multicentre study
>
> [5] ADformer: A Multi-Granularity Spatial-Temporal Transformer for EEG-Based Alzheimer Detection

---

### Meta-Review · Area_Chair_mtHs · 2025-12-22

**Summary:**

- Most reviewers raised concerns about the methodological novelty and depth, emphasizing that most contributions seem to address data curation/handing/processing.

- Unclear clinical improvements.

- Potentially misleading use of the term "foundation model"

- Open questions about data (pre-)processing, such as  spatial interpolation of channels, weakening the  neurophysiological interpretation.

- Questions about the evaluation procedure (leave-one-subject-out vs  subject-independent cross validation)

- Weak performance for early detection (such as predicting MCI at baseline).

**Reviewer Concerns:**

In my opinion, the unclear/lacking methodological novelty is indeed a very valid and important point of criticism. In their rebuttal, the authors argue that the contribution is more on the applied side (which is perfectly fine in general), but then there are also many open questions regarding applied aspects such as clinical improvements, early detection (which would be clinically relevant) and  neurophysiological interpretation, which could not be addressed by the authors is a fully convincing way.

**Reviewer Scores:**

I don't think that any of the reviewers would have changes their score significantly.

---

### Decision · Program_Chairs · 2026-01-26

Reject